# Preparation and Properties of Three Plasticiser-Free Novel Di-benzo-18-Crown-6 Aldimine-Derived Lead(II) Ion-Selective Electrodes

Deneikah T. Jackson [1], Peter N. Nelson [1,*], Kimberly Weston [2] and Richard A. Taylor [2]

[1] Department of Chemistry, The University of the West Indies Mona, Kingston 7, St. Andrew, Jamaica
[2] Department of Chemistry, The University of the West Indies, St. Augustine 685509, Trinidad and Tobago
* Correspondence: peter.nelson02@uwimona.edu.jm; Tel.: +1876-927-1910 or +1876-469-6675

**Abstract:** Three novel dibenzo-18-crown-6 aldimines were successfully synthesised and structurally characterised via various spectroscopic methods ($^{1}$H,$^{13}$H NMR, FT-IR) and their solution phase lead binding behaviours probed via absorption spectroscopy, the results are supported by Density Functional Theoretical (DFT) modelling. These methods revealed that the asymmetric nature of these compounds is such that at equilibrium the ether cavity adopts an open configuration where the constituent oxygen atoms exhibit a highly negative electrostatic potential; hence, they spontaneously ($\Delta G \sim -58$ kJ mol$^{-1}$) interact/bind aqueous lead ions to form stable 2:1 metal–ligand complexes. As indicated by cyclic and square voltammetry studies, all compounds are redox active and polymerise relatively easily onto a platinum surface to form a multi-layered lead Ion-selective Membrane (ISM), the structure of which is confirmed by Scanning Electron Microscopy (SEM) and Electrochemical Impedance Spectroscopy (EIS). These novel Ion-selective Electrodes (ISEs), as characterised by Differential Pulse Anodic Stripping Voltammetry (D PASV), allow selective electrochemical detection and quantification of lead at concentrations as low as 10 ppm, over a range of 15–60 ppm, with only minimal interference from mercury(II) and aluminium(III) ions at a 1:1 analyte-interferent ratio.

**Keywords:** imine; DFT; lead; dibenzo-18-crown-6; ion-selective electrode

## 1. Introduction

The biological toxicity of lead is known to occur through various mechanisms, one of which is its ability to mimic calcium ions, thereby interrupting cellular calcium channels which are responsible for cell function regulation [1]. Additionally, at the neuronal level, exposure to excess lead has been found to alter the release of neurotransmitters from presynaptic nerve endings, thereby interrupting various neurologically controlled processes [2]. Unfortunately, though these effects are well-known, increased mining activities in search of various "precious" metals for technological and other applications continue to increase the risk of human exposure to excess lead through various mechanisms: the contamination of groundwater, air pollution, and even agricultural produce (crops and animals), for example [3–5]. Furthermore, the continued application of lead in car batteries, solar and other technologies, coupled with the improper disposal of such energy storage and light harvesting equipment, represent a source of human exposure to such toxic species. Additionally, the application of lead in special lenses (binoculars, telescopes), glasses, lead pipes, and also the presence of lead-based paints in old buildings continue to guarantee human exposure to such toxic species on a continuous basis. Indeed, the continued exposure of humans to high levels of lead is highlighted by the recent Flint water crisis in Michigan USA, where over one-hundred thousand residents were exposed to dangerous levels of lead [6]. Similarly, residual lead from leaded fuels, though such fuels have been banned since the late 1990s, has been found to account for ca. 32–43% of lead in airborne particles in the United Kingdom [7], a clear indication of the environmental persistence

of lead. Such issues indicate the urgent need for fast, cheap and accurate analytic tools for investigating lead concentration in various media. Hence, numerous researchers have dedicated significant research efforts towards the development of electrochemical [8–10] as well as colourimetric sensors for the detection and quantification of lead. However, whereas colourimetric sensors offer the prospect of being applied in paper technology, thereby allowing "naked-eye" detection, such sensors are often restricted by selectivity, sensitivity, and stability issues. Alternatively, electrochemical sensors offer reliability, sensitivity, selectivity and scalability. Such systems make use of Ion-selective Electrodes (ISE) as the detection interface, allowing for metal selection and also relatively easy modification to facilitate the detection of other analytes in different sample matrices. Such features of ISEs are generally afforded by the application of an appropriate Ion-selective Membrane (ISM), hence, it is critical that fundamental knowledge of the interactions between the analyte and the ISM is established if high-quality pre-planned ISEs are to be prepared. These membranes are usually derived from molecular systems such as cryptands, thia-crowns, crown ethers and even pincer-type systems [9,11–13] which are all capable of forming strong coordination bonds or Host-guest complexes with various metal ions. However, oxo-crown systems are of particular interest since they are known to be highly stable and are generally inert. In particular, benzo-crowns offer the possibility of being easily tailored along with the ability to interact with metal ions through size binding and other effects [14,15]. In fact, previously, we have proven that benzo-crowns can interact strongly with lead and can be easily modified to yield compounds which interact differently with $Pb^{2+}$ under aqueous conditions. We have also proven that they can be readily polymerised to yield modified electrodes which can be applied in the detection and quantification of lead [8].

Hence, herein, a thorough exploratory study of the structural, spectroscopic, electrochemical, and lead binding behaviour of three novel dibenzo-18-crown-6 imines, synthesised as depicted in Scheme 1, is presented to establish a fundamental understanding of their binding, redox, spectroscopic and structural properties. Molecular modelling calculations, based on DFT methods, are also applied in elucidating the origins of the experimentally observed chemical properties of these compounds. Their electropolymerisation to yield plasticiser-free ISEs is also presented and the resulting ISEs are characterised via electrochemical impedance spectroscopy (EIS) and scanning electron microscopy (SEM). The analytical parameters of these ISEs, where the detection and quantification of lead is concerned, are also presented and compared to a well-regarded traditional method. Overall, the aim of this investigation is to assess the application potential of these imines for the development of modified electrodes which can allow relatively cheap and rapid electrochemical sensing and quantification of aqueous lead ions without the requirement for labourious and expensive methods and expertise.

**Scheme 1.** General synthetic scheme for 4,4-Diformyl(2-aminophenol)dibenzo-18-crown-6 (DBAP), 4,4-Diformyl(2-amino-5-methylphenol)dibenzo-18-crown-6 (DBMAP) and 4,4-Diformyl(2-amino-5-nitrophenol)dibenzo-18-crown-6 (DBNAP). TFAA (trifluoroacetic acid), HMTA (hexamethylenetetramine).

## 2. Discussion

### 2.1. Structure Optimisation and Infrared Spectroscopy

For a fundamental understanding of the molecular properties of the titled compounds, DFT calculations were carried out in order to locate their most probable molecular conformation, the correctness of which is confirmed by the absence of negative Eigenvalues in the Hessian. These results reveal that all structures adopt a non-planar geometry at the global energy minima, as expected; that is, the phenyl rings as well as the ether cavity, adopt a "V-shaped" conformation where the ether cavity occupies the apex of this V-shaped structure, with the phenyl ring portions arranged at an angle of ca. 57°, relative to each other, for all compounds (Figure 1b). Indeed, such a conformation was also reported elsewhere [8,15,16] for analogous compounds. Interestingly, for all three compounds, the rings derived from 2-aminophenol, hereafter referred to as the "I-ring", are arranged differently on either side of the ether cavity; that is, whereas on one side of this cavity, the I-ring is coplanar with the adjoining dibenzo ring (DB-ring), the other adopts a nearly perpendicular conformation relative to its adjoining DB-ring (Figure 1). This indicates an overall asymmetric molecular electron density in all three compounds since the co-planarity of these rings affects orbital overlap and hence, electron density distribution; that is, coplanar rings support greater electron density delocalisation. Of course, in order to confirm this hypothesis, additional data such as molecular orbital coefficients/distribution or electrostatic potential maps (ESPMs) are required (vide infra).

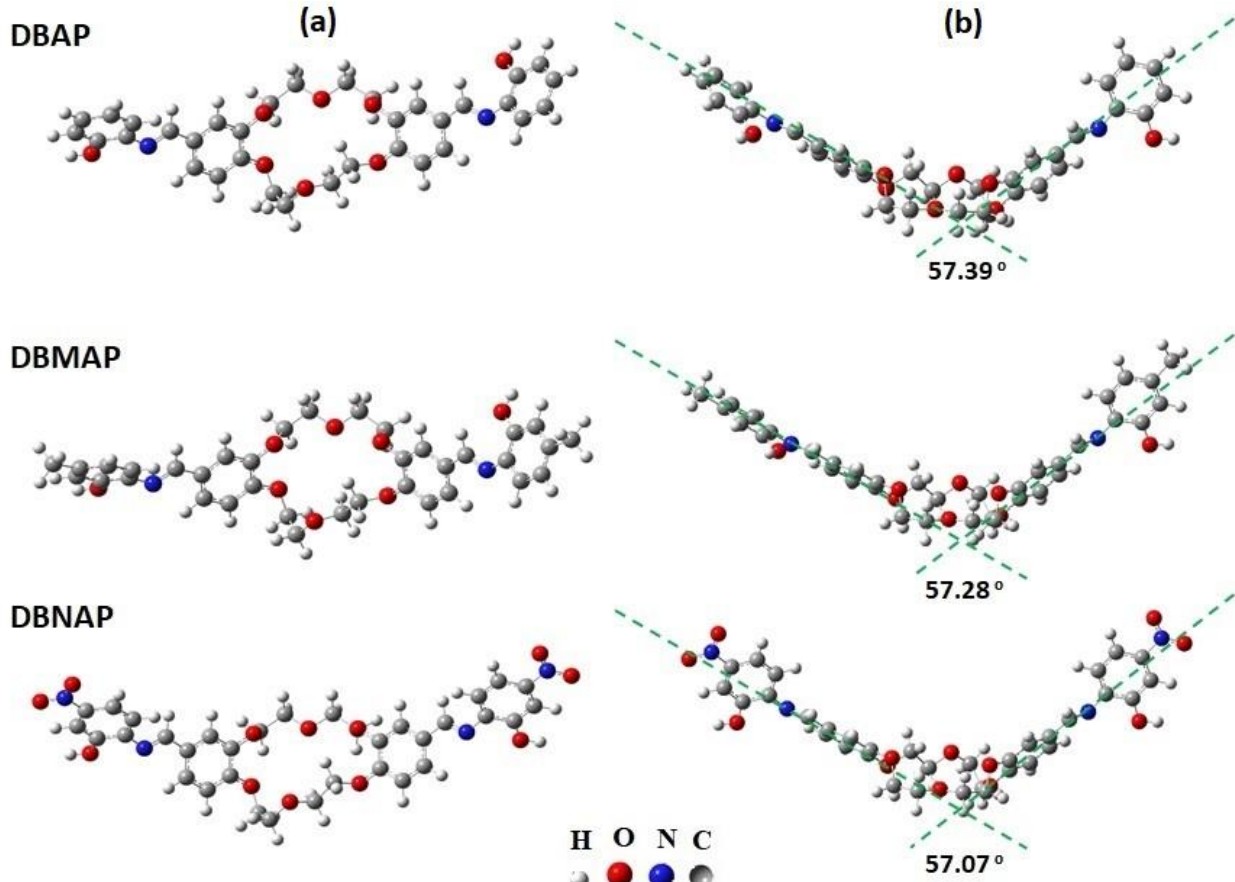

**Figure 1.** (**a**) Aerial view showing the open cavity arrangement and (**b**) side view of the optimised structures for 4,4-Diformyl(2-aminophenol)dibenzo-18-crown-6 (DBAP), 4,4-Diformyl(2-amino-5-methylphenol)dibenzo-18-crown-6 (DBMAP) and 4,4-Diformyl(2-amino-5-nitrophenol)dibenzo-18-crown-6 (DBNAP) revealing the "V-shaped" conformation of the overall structure which aids coordination of lead with the ether cavity. All structures were optimised based on DFT/6-311++ G(d,p)/RB3LYP.

Interestingly, for all compounds, the molecular conformation is such that the ether cavity adopts an open arrangement, the result of different configurations of the chains that make up the ether ring; that is, in one chain, the -CH$_2$-**CH$_2$-O-CH$_2$**-CH$_2$- hydrogens are eclipsed whereas in the other, this same fragment (highlighted) consists of methylene groups in a gauche conformation.

For a further structural understanding of these compounds, their bond distances were also calculated. For instance, C=N distances were calculated for all compounds in the range of 1.279–1.281 Å, values similar to those reported in the literature, based on X-ray crystallographic data [17,18]. However, for a given compound, these values are not identical for both sides of the molecule. Nonetheless, this distance does not vary much across all three compounds; that is, they are basically identical, the slight differences most likely originating from different orientations of the I-rings. Calculated carbon–carbon distances are also similar to those reported in the literature [19,20], based on X-ray data. For instance, C-C distances in both I-rings are in the range of 1.414–1.389 Å, whereas for the DB-rings, these distances are calculated between 1.405 and 1.388 Å. However, it should be noted that these distances are longest for bonds directly attached to carbons bearing non-hydrogen substituents, indicating ring deformation/elongation and hence, reduced phenyl ring symmetry. Interestingly, for all three compounds, the order of the C-O bond which anchors the ether chains to the DB rings is 1.5, indicating partial conjugation of the oxygen-based *n*-electrons with the phenyl π-system. This also indicates that phenyl ring substitution will have significant effects on the electron density and hence, metal coordination behaviour of the ether cavity. Based on this hypothesis and the fact that these C-O distances are shortest for DBNAP, it is reasonable to postulate that the I-rings have the most significant influence on the electronics of the ether cavity in this compound. As is typical, aromatic C-H distances are calculated in the range of 1.085–1.083 Å whereas the aliphatic ether C-H distances are in the range of 1.099–1.102 Å [21,22]. The distances for all other bonds, collected in Table S1a–c, also show marked similarities to experimental results, a clear indication that the model chemistry applied in the optimisation of these compounds is of sufficient quality to facilitate meaningful conclusions regarding the fundamental properties of the titled compounds. However, for a greater fundamental understanding of their properties, additional data are required. Hence, solid-state infrared spectra were collected for all compounds, at ambient temperature, revealing medium (m) to strong (s) bands in the low-frequency region of 1700–600 cm$^{-1}$ and weak (w) bands in the high-frequency region of 2800–3100 cm$^{-1}$ (Figure 2). The bands in the high-frequency sub-region between 2933 and 2824 cm$^{-1}$, exist as a cluster of transmissions, most likely associated with the ether C-H symmetric ($\nu_{ss}$) and asymmetric ($\nu_{as}$) stretching modes of the $\alpha$-CH$_3$ group, in addition to those for the CH$_3$ moiety of DBMAP. The presence of multiple bands in this sub-region indicates that the ether chains are conformational asymmetric; that is, the chain segments are non-equivalent, as indicated by the optimised structures (vide supra). Furthermore, as is typical [23], the aromatic $\nu_s$C-H vibrations are observed as a weak cluster of broad bands between ca. 3100 and 3063 cm$^{-1}$, for all compounds. The broadness of these bands is due to the presence of multiple closely spaced vibrations emanating from the overall low molecular symmetry which leads to disruption in the vibrational degeneracy of the aromatic C-H stretching modes. Such complexity offers a tremendous challenge in understanding the vibrational behaviour of these compounds from a fundamental perspective, hence, harmonic spectra for all compounds are calculated. Interestingly, though harmonic and calculated are based on an isolated molecule, the calculated spectra are highly similar to those derived experimentally, hence, they are discussed, herein, without scaling, further confirmation that the model chemistry applied in this work is of appropriate quality to allow accurate understanding of the infrared features of these compounds.

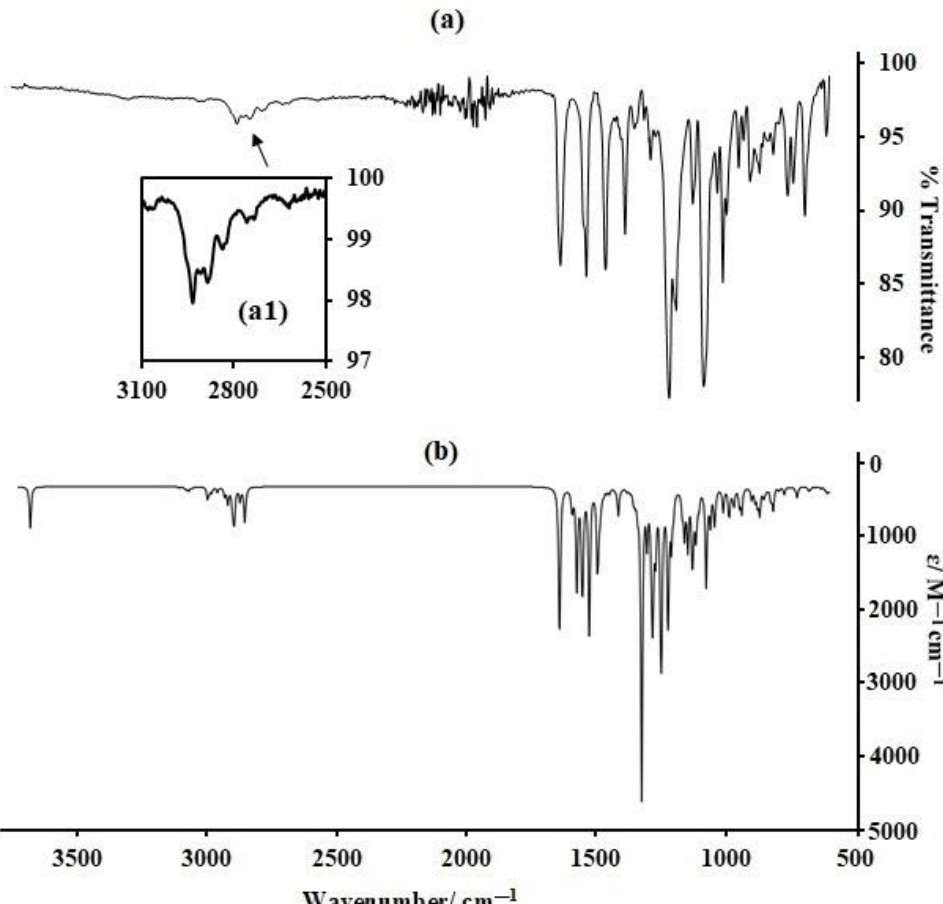

**Figure 2.** (**a**) Experimental solid-state infrared spectra and (**b**) calculated spectra for 4,4-Diformyl(2-amino-5-nitrophenol)dibenzo-18-crown-6 (DBNAP), as adduced from the harmonic oscillator model, calculated based on DFT/6-311++G(d,p)/RB3LYP in the gas phase.

Hence, vibrational animation of the calculated spectra is applied in assigning the experimentally observed vibrations. For instance, the bands observed experimentally between 2835 and 2831 cm$^{-1}$, corresponding to those calculated between 2955 and 2951 cm$^{-1}$ are due to the $\nu_{ss}$C-H vibration of the ether chains. However, the $\nu_{as}$C-H vibrations are calculated as multiple bands between 3112 and 3043 cm$^{-1}$. Of course, whereas the high stretching frequency of the $\nu_{as}$C-H is typical [24–26], the presence of multiple bands for this single mode is a result of disruption in the vibrational degeneracy among the CH$_2$ assembly of the ether chains. This region also consists of the aldehydic C-H stretching mode, calculated at ca. 3004 cm$^{-1}$ as two closely spaced bands, separated by less than ~0.5 cm$^{-1}$, due to differences in the aldehydic C-H environments on both sides of the ether ring. As is typical [27], the aromatic C-H stretches are calculated in the region of 3200–3172 cm$^{-1}$ as a cluster of low-intensity bands, similar to that observed experimentally. However, the slightly higher frequencies calculated for this mode might be due to the neglect of intermolecular interactions in these calculations.

The low-frequency region of 1700–600 cm$^{-1}$ is composed of several highly important bands (Figure 3). For instance, the $\nu_{s}$C=N vibration is calculated as two closely spaced ($\Delta\nu \approx 8$ cm$^{-1}$) bands between 1691 and 1679 cm$^{-1}$ but experimentally observed between 1686 and 1672 cm$^{-1}$ as a broad (Full Width and Half Maximum, FWHM: 46 cm$^{-1}$) strong band for, all compounds.

Such observations indicate that the experimental C=N band is most likely composed of more than one closely spaced vibration due to the low overall molecular symmetry of these compounds, hence, the disruption in the vibrational degeneracy of both C=N moieties. This low-frequency region also consists of the aromatic C=C stretching vibrations which

are calculated as multiple bands between 1646 and 1595 cm$^{-1}$ but are observed experimentally as a broadband (FWHM: ~65–74 cm$^{-1}$) at 1586 cm$^{-1}$ with a conspicuous shoulder at 1596 cm$^{-1}$, for all compounds. Interestingly, animation of the calculated spectra reveals that the C=C moieties for all four rings vibrate at slightly different frequencies, hence, multiple bands are calculated for this same mode, a clear explanation for the broadness of the experimental band. For DBNAP, in addition to the $\nu_s$N=O at ~1520 cm$^{-1}$, two other bands are calculated at ~1537 and 1450 cm$^{-1}$ (Figure 3a,b) due to the combined ether ring CH$_2$ and the aromatic C-H rocking motions. It should be noted that different regions of the ether ring as well as the different aromatic C-Hs, rock at different frequencies, hence, both the calculated and experimentally observed bands are highly asymmetric in shape (Figure 3). However, the bands in the neighbouring lower energy region represent a combination of several vibrational modes: combination bands, as indicated by the animated harmonic spectra. Overall, these results indicate that all three compounds are similar in terms of their vibrational behaviour and low symmetry but are not identical. However, in order to gain deeper insights into their electronic structure and lead-binding behaviour, additional data are required.

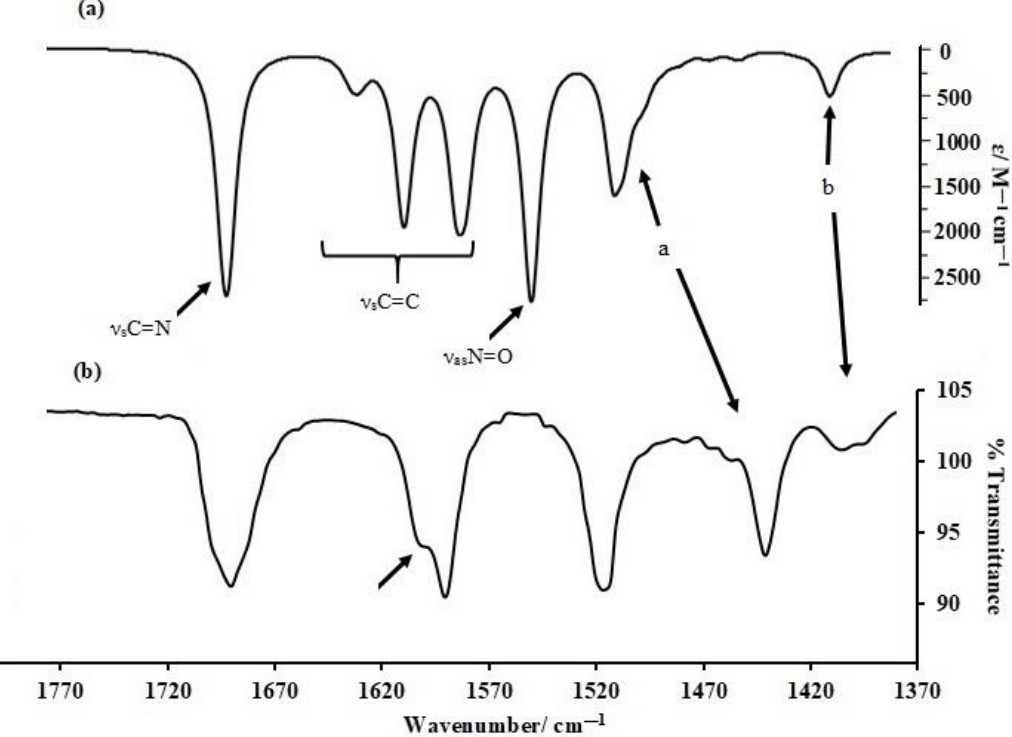

**Figure 3.** (**a**) Calculated and (**b**) experimental low-frequency IR spectral region, exemplified by 4,4-Diformyl(2-amino-5-nitrophenol)dibenzo-18-crown-6 (DBNAP). The calculated spectra are derived from the harmonic oscillator model and calculated here via DFT/6-311++G(d,p)/RB3LYP.

## 2.2. Absorption Spectroscopy

Absorption spectra for all three compounds, collected at ambient temperature in spectroscopy grade DMSO (Sigma, 99.9%), revealed at least two absorptions for all compounds (Figure 4). For instance, the spectrum for DBMAP revealed two absorptions, one at 282 and the other at 308 nm along with what appears to be a broad weak shoulder at ca. 380 nm. Similarly, for DBNAP, a broad low energy band is observed at ca. 418 nm, probably due to the forbidden n to π* transitions in the imine moiety, hence, it is of relatively low probability. However, the broadness of this absorption: FWHM ≈ 66 nm, indicates that it is most likely composed of multiple or at least two closely spaced absorptions which, themselves, might be composed of multiple transitions. However, the higher energy bands for this same compound, observed at 280 and 314 nm, are most likely associated with

π to π* transitions, probably between the DB- and the I-ring's π-system or even the imine pi-system. For DBAP, only one obvious high-energy absorption is observed at ca. 281 nm with two low-energy shoulders at ca. 308 and 355 nm (Figure 4 (*)). Additionally, a broad low energy band is observed at ca. 430 nm, similar to that observed for the two other derivatives, however, this absorption shows two clear "humps", indicating that it is composed of at least two closely spaced peaks. Such observations probably indicate that the C=N moieties on both sides of the ether ring are of sufficient electronic difference that they absorb at different wavelengths, the same is probably true for the phenyl π-systems on both sides of the ether cavity. Additionally, disruption in the degeneracy of the constituent transitions of these absorptions might be facilitated by the low overall molecular symmetry, hence, more absorptions than expected are observed. Interestingly, for all compounds, the high energy absorption is most probable, indicating that they are associated with transitions of the highest dipole moment and hence, highest oscillator strengths. However, for DB-NAP, two clear bands of similar oscillator strength are observed between 255 and 310 nm, which are most likely associated with π to π* transitions. Of course, the resolution of these two absorptions, relative to the other two derivatives, suggest noteworthy differences in their energetics at the global energy minima.

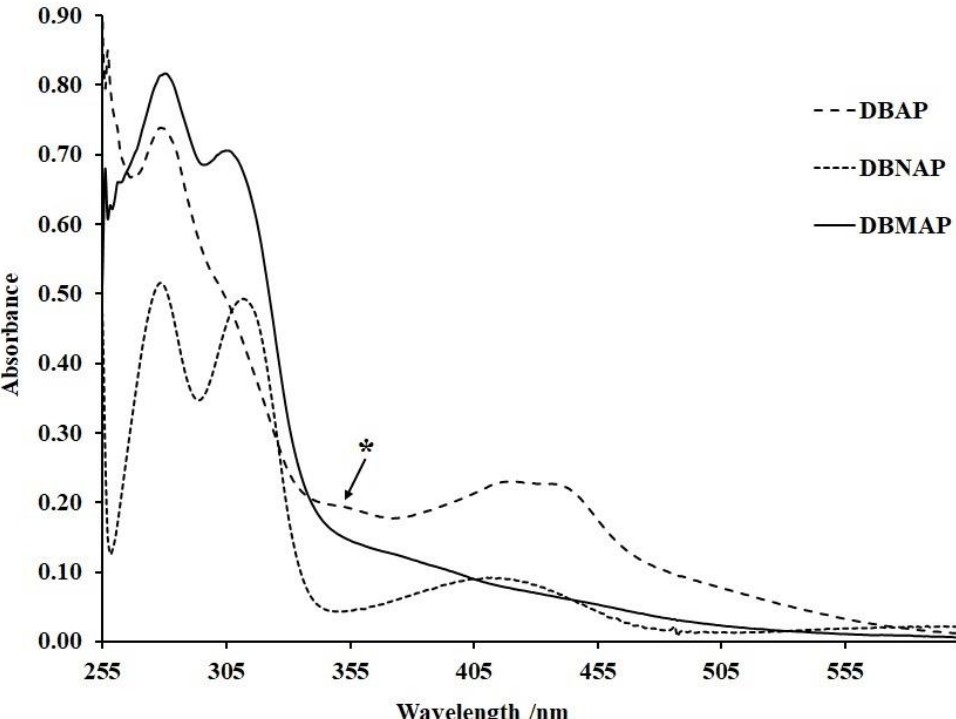

**Figure 4.** Experimental absorption spectra for 4,4-Diformyl(2-aminophenol)dibenzo-18-crown-6 (**DBAP**), 4,4-Diformyl(2-amino-5-methylphenol)dibenzo-18-crown-6 (DBMAP) and 4,4-Diformyl(2-amino-5-nitrophenol)dibenzo-18-crown-6 (DBNAP) (Conc. $1.0 \times 10^{-4}$ M) at 298.15 K and 1 atm, collected in DMSO. These spectra reveal numerous strong bands in addition to various low energy absorptions (*).

Given the complexity of these absorption spectra, time-dependent density functional theoretical calculations (TD-DFT), using the aforementioned modelling parameters, were conducted to aid such characterisation. Indeed, the calculated spectra reveal three main absorptions in all cases, associated with the migration of electron density to and from various molecular regions. For example, in the case of DBAP, the highest energy absorption, calculated at 328 nm, is composed of a single transition: HOMO to LUMO (99.5%), emanating from electron density migration from the aromatic and imine π-system on one side of the molecule to the π* orbitals on the opposite side of the molecule (Figure 5). The energetic similarity between this transition and that observed experimentally in the

vicinity of 315 nm, for all compounds, indicates their correspondence. However, two other absorptions are calculated at 358 and 361 nm. The former is associated with a HOMO-1 to LUMO (6.5%) transition, emanating from π to π* electron density migration on one side of the molecule, in addition to a HOMO to LUMO + 1 (84.5%) transition, due to π to π* electron density migration between the *I*- and DB-rings on one side of the ether cavity (Figure 5). However, the latter is composed of a single transition: HOMO-1 to LUMO (86.8%), due to π to π* electron density migration, similar to that calculated for the band at 358 nm.

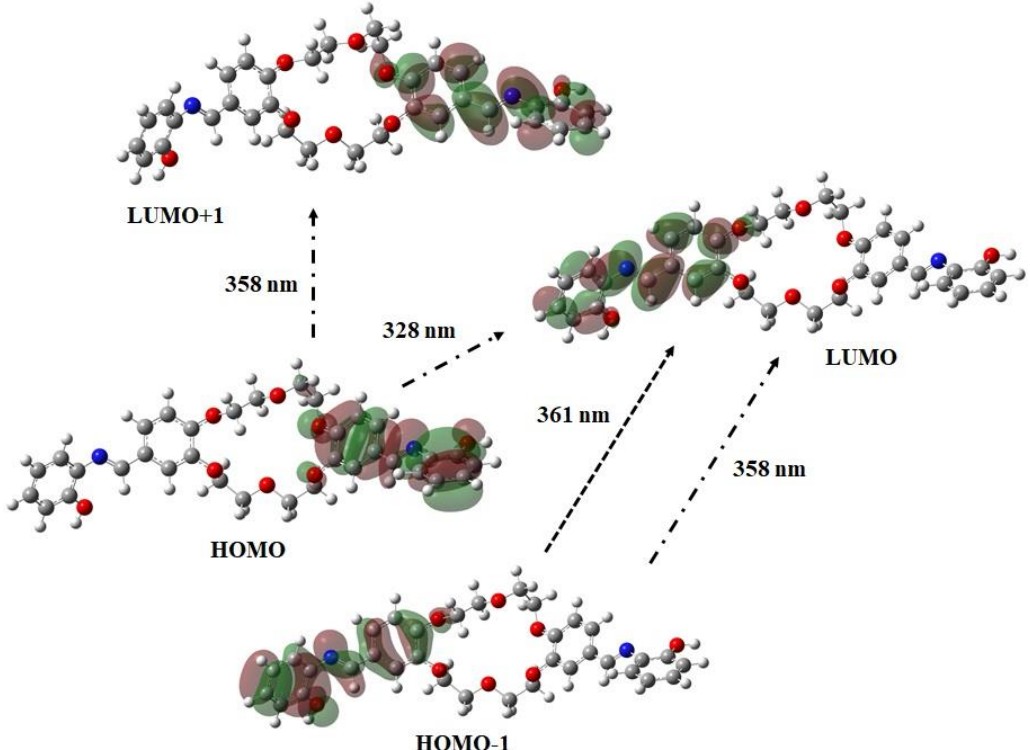

**Figure 5.** Calculated molecular orbital surfaces, exemplified by 4,4-Diformyl(2-aminophenol)dibenzo-18-crown-6 (DBAP). The calculations were conducted in acetonitrile where solvent effects are correct for using the C-PCM model and structure optimized based on TD-DFT/6-311++G(d,p)/RB3LYP.

However, for DBNAP, where a single broad absorption is observed at ca. 418 nm, as outlined in the foregoing, two closely spaced absorptions are calculated in this region: 418.2 and 416 nm, consisting of a single transition in both cases: HOMO-1 to LUMO (94.1%) and HOMO to LUMO (98.0%), respectively. Of course, the closeness of these two transitions confirms that the broadness of the experimentally observed band is due to the presence of two closely spaced absorptions. However, though these bands are close, the calculated MO surfaces show that the HOMO-1 to LUMO transition involves π to π* electron density migration from the aromatic rings on one side of the molecule to those on the other side; that is, the HOMO-1 is concentrated both on the I- and DB-rings, whereas the LUMO has its largest coefficient on the I-ring and the imine moiety of the same side of the molecule (see Table S2). However, the HOMO to LUMO transition, calculated at 416 nm, results from charge transfer from one side of the molecule (DB + I-ring) to the I-ring on the other side of the molecule. Interestingly, an additional peak is calculated at 442.5 nm, associated with the HOMO to LUMO + 1 (98.0%) transition resulting from electron density rearrangement on one side of the molecule; that is, between the DB and I-ring (see Table S2). However, despite the closeness of the calculated and experimental bands, the highest energy, observed experimentally (vide supra), was not predicted by these calculations. The divergence between the calculated and experimental results might be due to slight differences in the global energy minima as calculated versus that existing under experimental conditions.

Indeed, this might be the result of intra- and or inter-molecular hydrogen bonding effects. Additionally, the absence of explicit solvation effects, features not accurately predicted by the applied implicit solvation model might also be implicated in such divergence. Similar results are also observed for DBMAP; that is, three absorptions are calculated at 334, 364 and 366 nm. However, the band calculated at the highest energy is energetically similar to that observed experimentally at 308 nm. As calculated, this absorption is composed solely of the HOMO to LUMO (99.6%) transition, associated with electron density migration from one side of the molecule to the other, inclusive of the imine moiety; that is, this transition is composed of a combination n to $\pi^*$ and $\pi$ to $\pi^*$ electronic rearrangements (see Table S2). However, the absorption, calculated at 366 nm but experimentally observed at ca. 379 nm, is composed of two transitions: HOMO to LUMO + 1 (59.6%) and HOMO-1 to LUMO (33.6%), the former being associated with $\pi$ to $\pi^*$ electron density migration from one side of the molecule to the other, whereas the latter is due to $\pi$ to $\pi^*$ electron density rearrangement on one side of the molecule, (see Table S3). Such similarities between the compositions of these absorptions indicate significant electronic likeness between these compounds, however, differences in the contribution of the various molecular orbitals involved in each transition could have tremendous effects on their interactions with lead (II) ions and hence, their application potential in the preparation of high sensitivity and selectivity lead sensors.

Hence, lead (II) ions were added to individual solutions of the synthesised compounds in a stepwise manner and the absorption spectra were collected after each addition: Uv-vis titration. Such step-wise addition of lead (II) resulted in increased intensity of the bands at lowest energy (Figure 6a), for all compounds, indicating that they form strong interactions with lead ions. Furthermore, a plot of absorbance versus metal–ligand ratio indicates the formation of a 1:1 complex which later evolves into a more stable 2:1 system (Figure 6b), as expected, given the presence of the two imine moieties and the 2-hydroxy (2-OH) groups on the I-rings, in addition to the ether cavity. Furthermore, given the possibility of binding at these three sites, titration beyond a 3:1 metal-to-ligand ratio was explored but no inflexion point was observed. This indicates that coordination at two of the three binding sites prevented coordination at the third, probably through reduced electron density at the third site or steric hindrance caused by the new structural conformation resulting from binding. Unfortunately, attempts to grow crystals of the lead complexes of these compounds for X-ray diffraction were unsuccessful.

Nonetheless, infrared spectral measurements on powdered samples of these complexes reveal several noteworthy changes. For example, shifting of the $\nu_s$C=N vibration to lower frequency with reduced intensity, a clear indication of reduced bond order, confirms coordination at this site. Furthermore, for all compounds, coalescence of the bands between 1500 and 1400 $cm^{-1}$, ascribed to the ether $CH_2$ rocking motion, as outlined in the foregoing, suggests coordination with the ether cavity (see Figure S1a–c). Indeed, optimisation of the lead complex, for all compounds, reveals a shift of the imine C=N vibration to a lower frequency with reduced relative intensity. Additionally, the calculated spectra for the complex reveal increased energy separation between the two imine moieties, since one is involved in metal coordination whilst the other is free. Furthermore, similar to experimental observations, reduced separation between the $CH_2$ rocking modes is calculated. Moreover, coordination at these two sites is supported by the calculated electrostatic potential maps (Figure 7) which reveal high relative electron density in the region between the 2-OH and the imine moieties which are coplanar with the *I-ring*, hence, the 2-OH and the C=N binding sites are syn-periplanar.

Similarly, the high electron density calculated for the ether cavity suggests possible spontaneous coordination with lead ions. Therefore, the majority of evidence is for coordination with one of the imines and 2-OH moieties and the ether cavity, yielding a 2:1 metal–ligand complex. For a greater understanding of the interactions of these systems with lead (II), their binding/association constants (Table 1) are calculated via the application

of Equation (1), as presented and successfully applied by Nonaka and Hamada [28] to yield Figure 6c,

$$\log K = \log[(A_o - A_{eq})/(A_{eq} - A_\infty)] - n \log[M] \tag{1}$$

where K is the association constant, n is the binding ratio, M is the concentration of the metal ion (guest) and $A_o$, $A_{eq}$ and $A_\infty$ represent the initial, equilibrium and infinity absorbance values of the metal–ligand solution. Surprisingly, the binding constants for all three compounds are highly similar (Table 1), indicating that their metal–ligand binding strengths and structures are basically identical. Indeed, this proposal is confirmed by their binding free energies which, of course, are basically identical. Nonetheless, all binding free energy values are exergonic, indicating spontaneous binding of lead (II).

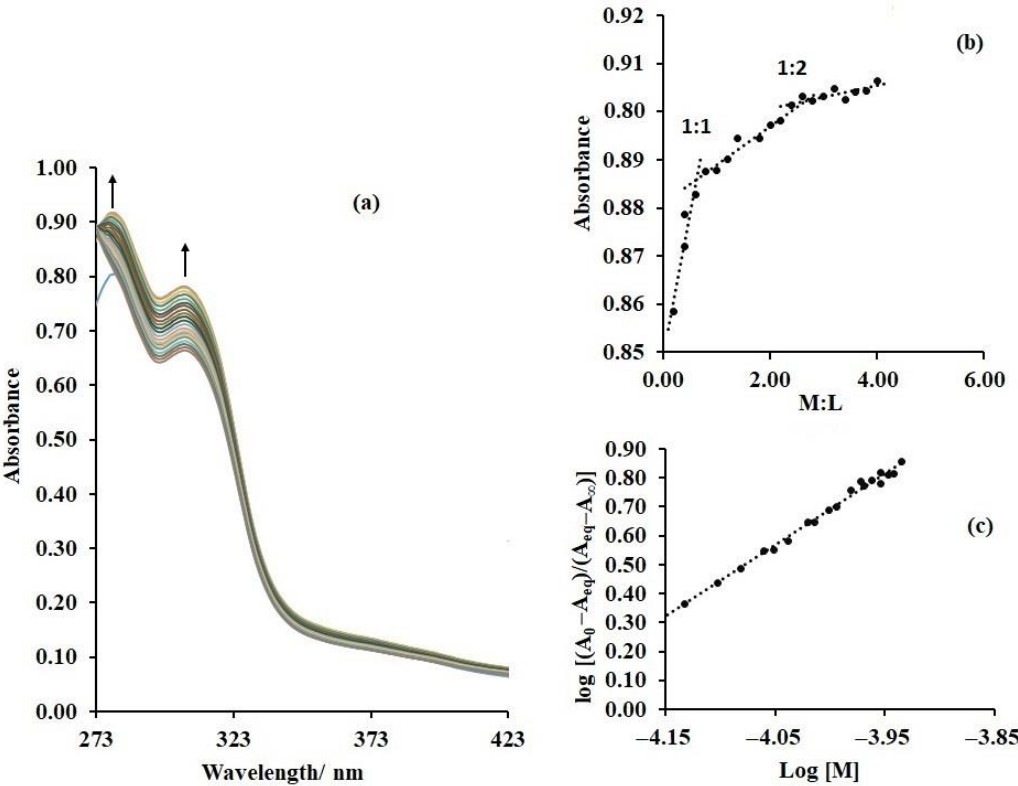

**Figure 6.** (**a**) Optical absorption spectra in the presence of different concentrations of lead ($[Pb^{2+}]$ = 100 μM addition per-step 60 μL). (**b**) Mole ratio and (**c**) Hill plot, exemplified by 4,4-Diformyl(2-amino-5-methylphenol)dibenzo-18-crown-6 (DBMAP).

**Table 1.** Binding thermodynamic parameters for 4,4-Diformyl(2-aminophenol)dibenzo-18-crown-6 (DBAP), 4,4-Diformyl(2-amino-5-methylphenol)dibenzo-18-crown-6 (DBMAP) and 4,4-Diformyl (2-amino-5-nitrophenol)dibenzo-18-crown-6 (DBNAP). ΔG values are determined from Boltzmann's model whereas ΔH and ΔS are determined from Van't Hoff's model.

| Compound | Log $K^{298.15\,K}$ | ΔG [kJ mol$^{-1}$] | ΔH/R [K$^{-1}$] | ΔS/R |
|---|---|---|---|---|
| DBAP | 10.29 ± 0.66 | −57.75 | 57,161.30 ± 4675.99 | 214.95 ± 15.69 |
| DBNAP | 10.20 ± 0.80 | −7.26 | 51,072.20 ± 10962.63 | 193.79 ± 36.80 |
| DBMAP | 10.54 ± 0.19 | −59.13 | 31,455.60 ± 6490.11 | 129.27 ± 21.78 |

Binding enthalpies and entropies, as determined (Table 1) based on the popular and well-regarded Van't Hoff's model: $\ln K = -\Delta H/RT + \Delta S/R$, reveal that binding is endothermic in all cases; however, increased energy disorder (+ΔS), caused by complexation, allows for favourable binding. Unfortunately, despite several attempts, the margin of error in the values of ΔH/R and ΔS/R does not allow the identification of any real binding

thermodynamic differences between all three compounds. However, given the differences in their electron density distribution and hence, polarity, their redox properties should reflect noteworthy differences. Nonetheless, since they all interact strongly with $Pb^{2+}$, their application potential/usefulness as lead ion sensors are indicated.

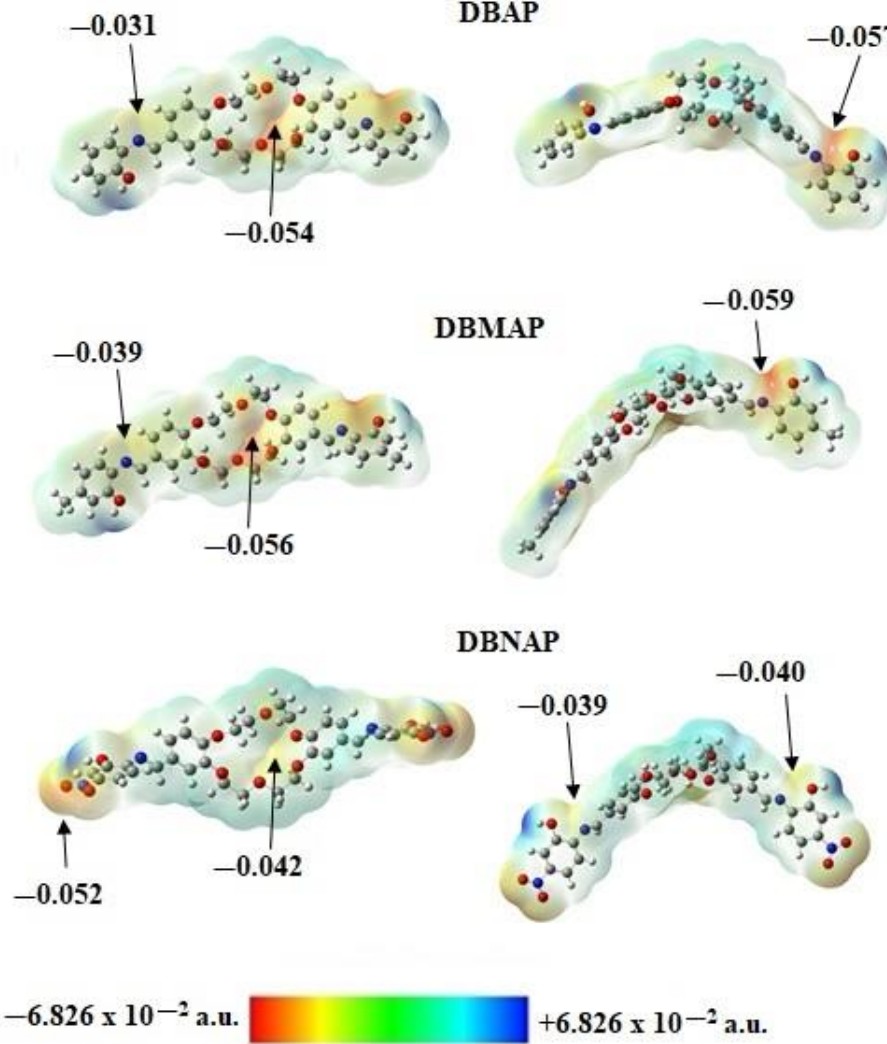

**Figure 7.** Areal and side views of the calculated Electrostatic potential map (ESPM) surfaces (Isoval. 0.0040) for 4,4-Diformyl(2-aminophenol)dibenzo-18-crown-6 (DBAP), 4,4-Diformyl(2-amino-5-methylphenol)dibenzo-18-crown-6 (DBMAP) and 4,4-Diformyl(2-amino-5-nitrophenol)dibenzo-18-crown-6 (DBNAP). The electrostatic potential maps were calculated using DFT/6-311++G(d,p)/RB3LYP.

*2.3. Redox Behaviour*

The redox behaviour of all three compounds was investigated via cyclic voltammetric measurements in Uv/HPLC grade acetonitrile (PHARMC-AAPER), a solvent in which the compounds are very soluble, with tetrabutylammonium hexafluorophate (Sigma, $[N(Bu)_4PF_6] = 0.05$ M) as supporting electrolyte, under a nitrogen atmosphere, within the potential window of $-2.5$ to $+2.5$ V versus an Ag/AgCl reference. Indeed, such a wide potential window further justifies the use of acetonitrile for this study. In all cases, a platinum working electrode (WE) and platinum wire counter electrode (CE) were used. Oxidatively initiated scans for all three compounds exhibit at least a single irreversible oxidation peak at ~1.45 V (Figure 8a) with no noticeable reductions in that vicinity even at high or low scan rates, indicating an electron transfer–chemical reaction mechanism: EC mechanism. Interestingly, whereas for DBMAP only a single irreversible oxidation is

observed at ca. 1.46 V, for DBNAP, a low current adsorption peak is also obvious on the low energy side of the main oxidation, at ca. 2.29 V. Similarly, for DBAP, two pre-oxidation peaks are observed between 0.7 and 1.48 V, whereas the main oxidation band is observed at 1.70 V. These pre-oxidation peaks originate from physical adsorption of the compounds onto the electrode surface before oxidation, a clear indication that they are likely to passivate the electrode surface on multiple scans. Surprisingly, reductively initiated scans, for all compounds are nearly identical to these voltammograms, both in terms of current and potential of the observed peaks, of course, with some minute differences. For instance, in the case of DBMAP (Figure 8), the main oxidation peak is shifted to slightly higher energy; however, this shift is so minimal: 1.46 to 1.56 V, no real conclusions can be drawn on this basis since even slight changes in electrode configuration could lead to such effects.

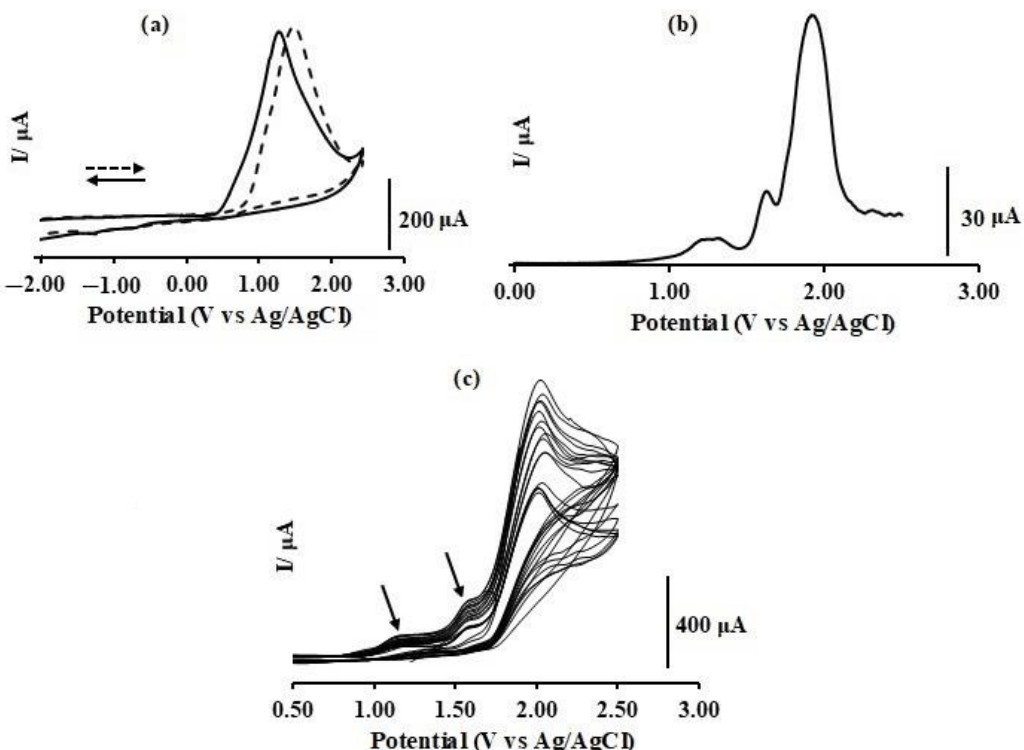

**Figure 8.** (**a**) Cyclic (scan rate (v) = 200 mV/s) and (**b**) square wave voltammogram for 4,4-Diformyl(2-amino-5-methylphenol)dibenzo-18-crown-6 (DBMAP), (**c**) multi-cycle (30 scans)/polymerisation scan, exemplified by 4,4-Diformyl(2-amino-5-nitrophenol)dibenzo-18-crown-6 (DBNAP). All data were collected under a nitrogen atmosphere in acetonitrile, using a three-electrode set-up with platinum working and counter electrodes. Data for the other compounds are collected in Figure S4.

Application of the popular Randles-Ševčík model for irreversible processes [29,30], represented by Equation (2), where $F$, $n$, $n'$, $A_{real}$, $D$ and $\alpha$ represent the Faraday's constant, the number of electrons transferred, the number of electrons transferred before the rate-determining step, the electroactive area, the diffusion coefficient and the transfer coefficient, respectively:

$$I_p^{irrev} = 0.496\sqrt{\alpha n'}\, nFA_{real}C\sqrt{\frac{nFDv}{RT}} \tag{2}$$

reveal a linear relationship between the current of the main oxidation and $v^{1/2}$ for all compounds. This is indicative of a diffusion-controlled process. Interestingly, when Log $I_p$ is plotted against Log $v$, as invoked by the law: Log $I_p$ = Log a + bLog $v$, b values of 0.9, 0.8 and 0.9 are calculated for DBAP, DBNAP and DBMAP, respectively, indicating that the associated mechanistic processes are both capacitive and diffusion controlled. However,

adsorption effects result in partial passivation of the electrode surface on each cycle, hence, cleaning of the electrode surface between cycles results in a reduced correlation coefficient ($R^2 = 0.87$ to 0.98) for these plots. Such observations confirm that the pre-oxidation bands, observed on the low energy side of the main oxidation peak, are in-fact due to adsorption effects, as postulated in the foregoing. Of course, for a greater fundamental understanding of the redox processes associated with these bands, square wave (SW) measurements were conducted for all compounds under nearly identical conditions. Comparison of these SW peaks to that for ferrocene, measured under similar conditions, reveals a two-electron process for DBAP, most likely due to the loss of one electron from each *I-ring*-based hydroxyl group where the molecular electron density is highest, as indicated by the calculated ESPM (Figure 7). However, for DBNAP, four (4) electrons are lost, probably from the $NO_2$ and the OH groups but for DBMAP three (3) electrons are lost, probability from the ether cavity and the I-ring OH groups, as suggested by the ESPM. Indeed, optimisation of the radical cations of DBNAP and DBAP, reveals significant depletion of electron density in regions of originally high electron density, as expected. Unfortunately, despite several attempts, optimisation of the radical cation for DBMAP was unsuccessful. Nonetheless, for DBAP and DBNAP, these calculations reveal significant changes in the overall molecular electron density, leading to an overall positive electrostatic potential. However, the most contrasting changes are observed for the ether ring, the I-ring-based hydroxyl and the nitro group in the case of DBNAP (see Figure S2). Additionally, increased acidity of the OH moieties due to oxidation of these compounds, indicates significant changes in the electronics of the OH moiety, most likely due to the loss of an electron. For further insights into the effects of oxidation on their electronics, Natural Bonding Orbital (NBO) calculations were conducted on both the radical cation ion and neutral compounds, revealing increased (more positive) natural charge on the OH oxygen and imine nitrogen atoms. However, this change was most significant for the OH moiety, indicating that the first electron was most likely lost from this function. Indeed, this is confirmed by Natural Population Analysis (NPA) which reveals reduced occupancy of the valence orbitals of the OH oxygen atoms. Furthermore, the fact that only minute changes are observed in the shared electron density and hybridisation of the arC-OH bond, the majority of evidence is for the loss of an *n*-electron from the OH moieties. The loss of electron density from the C=N and $NO_2$ moieties due to this first oxidation is probably due to electron density redistribution as the overall molecular conformation reorganises to yield the new global energy minima. As expected, analogous results were obtained for DBNAP. Overall, such results show that all three compounds are redox active and could probably be electropolymerised onto the WE or at least form a strong chemisorbed coating, thereon. Based on these results, the following structure (Scheme 2) is proposed for the resulting electro-polymer of these compounds, a structure similar to that generated from the oxidation of catechol, as reported elsewhere [31]:

**Scheme 2.** Oxidation-driven electro-polymerisation of 4,4-Diformyl(2-aminophenol)dibenzo-18-crown-6 (DBAP), 4,4-Diformyl(2-amino-5-methylphenol)dibenzo-18-crown-6 (DBMAP) and 4,4-Diformyl(2-amino-5-nitrophenol)dibenzo-18-crown-6 (DBNAP) monomeric species on the Pt electrode surface in acetonitrile at 25 °C. It should be noted that oxidation of DBNAP could lead to the transformation of the $NO_2$ moiety to NO(OH) or $N(OH)_2$ as is typical for such species.

Hence, repetitive consecutive voltammetric cycling (30 cycles), within the aforementioned scan window, was carried out for all compounds, a method typically used to effect electropolymerisation of an electroactive solute onto the WE [30]. The resulting voltammograms reveal marked changes in the main oxidation wave on each cycle; that is, a clear decrease in current on each cycle, concomitant with noticeable changes in peak potentials, is obvious (Figure 8c, see Figure S3a,b). Additionally, the emergence of new bands indicates that the observed electrode surface passivation is most likely caused by the formation of a polymeric film of the solution phase DB-derivatives, thereon. Of course, in probing the structure of this surface-bound film, the surface coverage ($\Gamma$) was evaluated based on Equation (3) since the reaction would have occurred on the surface subsequent to the adsorption of the solution phase species, as outlined in the foregoing:

$$ I_p = \frac{n^2 F^2 A \Gamma v}{4RT} \tag{3} $$

where $n$, $F$, $A$ and $T$ represent the number of electrons ($n = 1$ for Ferrocene), Faraday's constant, the original electrode surface area and the temperature, respectively. For DBMAP, a $\Gamma$ value of $1.60 \times 10^{-10}$ mol cm$^{-2}$ is calculated, a clear indication of monolayer coverage, according to Bard et al. [30,32]. However, for DBAP and DBNAP where $\Gamma$ is 2.13 and 3.73 ($\times 10^{-12}$) mol cm$^{-2}$, incomplete surface coverage is indicated, probably due to significant spacing of the polymeric films on the electrode surface or the presence of excess pores in the film. Additionally, such low $\Gamma$ values could also be indicative of the different molecular orientations of the surface-bound film, not accounted for in the surface coverage model, a feature which could offer, different interaction strengths/effects with the electrode surface. Furthermore, the formation of multilayer films where the orientation of the first layer is in the form of islands, is also a possibility.

Of course, in order to further probe the solid-state structure of these surface-bound films, electrochemical impedance spectroscopy measurements were carried out on all modified electrodes, of course, enclosed in a Faraday cage to reduce electrical noise. The resulting Nyquist plots for all three compounds are semi-circular (Figure 9a), with the smallest semi-circle being observed for DBNAP, indicating low charge transfer resistance ($R_{ct}$) relative to the other electrodes where, in addition to higher $R_{ct}$, greater ion diffusion and possibly greater capacitive behaviour is indicated. Indeed, Bode plots (Figure 9) for DBAP and DBMAP, reveal a larger capacitive region than that exhibited by DBNAP, a clear indication of greater surface roughness, probably caused by layering (multi-layer) or a granular structure, all factors contributing to surface inhomogeneities, which are most significant for DBNAP. Furthermore, the presence of two inflexion points in the phase plot for this electrode indicates two relaxation processes [33,34] that at high frequency (~100 Hz) are associated with the dielectric of the polymeric film, whereas that at low frequency (~1 Hz) emanate from the electrode surface which might or might not consist of a monolayer of the polymeric film. Such a thin film might be located at the base of a defect/pore (Figure 10b); the type of surface morphology and resulting impedance features were discussed in significant theoretical detail by Mulder [35].

In mapping the collected EIS data, the best fit was achieved via the equivalent electrical circuit (EEC) in Figure 10c, where the double-layer capacitance is replaced by a constant phase element (CPE) which accounts for the non-ideal capacitive behaviour of the polymeric film. The impedance for CPE is given by $Z_{CPE} = 1/[Y_o(j\omega)^n]$ where $Y_o$ is the admittance and when n = 1, purely capacitive behaviour is indicated. However, these parameters, collected in Table 2, show that n ~0.8 for all compounds, confirming excess surface roughness.

Of course, to confirm the conclusions drawn from EIS data, scanning electron microscopy (SEM) images were collected on the surfaces of all three modified electrodes. Firstly, these images (Figure 11) confirm that the electrodes are indeed coated with a film consisting of mounds and cracks. The images also reveal that the coatings are not monolayer as predicted in the foregoing, especially for DBNAP and DBAP. This means that these electrodes present larger interaction areas than that predicted by the surface coverage

model, discussed in the foregoing since they are actually three-dimensional (3D) rather than quasi-two-dimensional. Furthermore, the presence of cracks/irregularly shaped pores and mounds corroborate the non-ideal capacitive behaviour predicted by the EIS data.

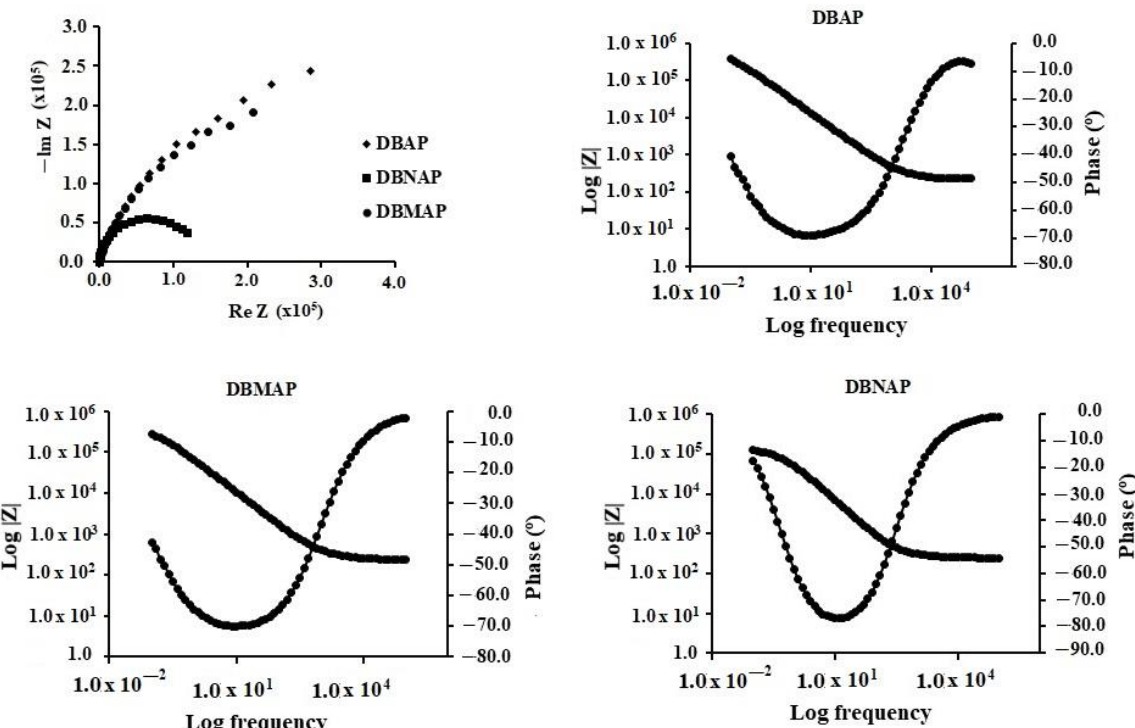

**Figure 9.** Combined Nyquist and individual Bode plots for 4,4-Diformyl(2-aminophenol)dibenzo-18-crown-6 (DBAP), 4,4-Diformyl(2-amino-5-methylphenol)dibenzo-18-crown-6 (DBMAP) and 4,4-Diformyl(2-amino-5-nitrophenol)dibenzo-18-crown-6 (DBNAP) derived ion-selective electrodes, indicating non-ideal capacitive behaviour in all cases.

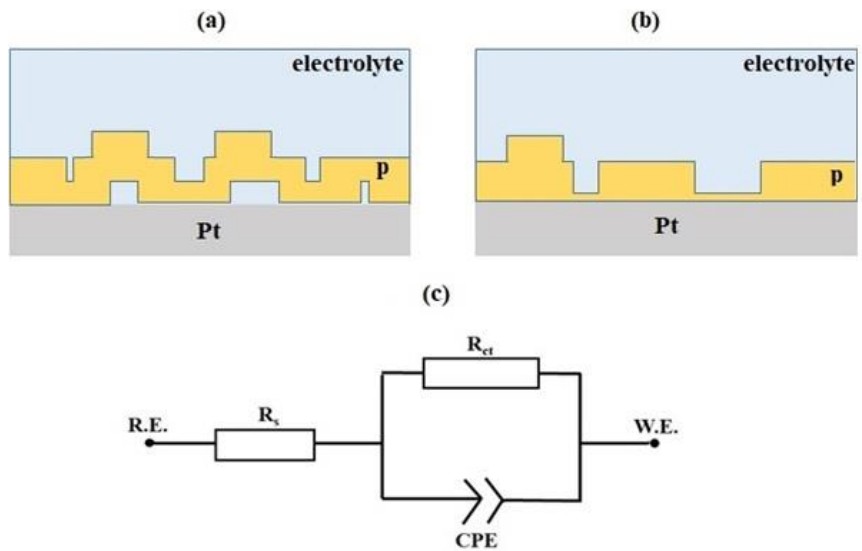

**Figure 10.** (**a,b**) Proposed multi-layer polymeric film structure (P) with different pore structure and (**c**) Equivalent circuit, providing the best fit for 4,4-Diformyl(2-aminophenol)dibenzo-18-crown-6 (DBAP), 4,4-Diformyl(2-amino-5-methylphenol)dibenzo-18-crown-6 (DBMAP) and 4,4-Diformyl(2-amino-5-nitrophenol)dibenzo-18-crown-6 (DBNAP). This model highlights the non-ideal capacitive behaviour of the polymeric ion-selective film/membrane on the electrode surface, indicating the presence of excess surface roughness and or pore in the polymeric film.

**Table 2.** Equivalent circuit parameters as derived for 4,4-Diformyl(2-aminophenol)dibenzo-18-crown-6 (DBAP), 4,4-Diformyl(2-amino-5-methylphenol)dibenzo-18-crown-6 (DBMAP) and 4,4-Diformyl(2-amino-5-nitrophenol)dibenzo-18-crown-6 (DBNAP) based on their impedance behaviour in acetonitrile, revealing that none of these ion-selective films offer ideal capacity behaviour.

| Compound | Yo [μS] | $R_s$ [Ω] | $R_{ct}$ [kΩ] | n |
|---|---|---|---|---|
| DBAP | 2.838 | 222 | 286 | 0.786 |
| DBMAP | 3.438 | 132 | 176 | 0.802 |
| DBNAP | 2.435 | 504 | 119 | 0.900 |

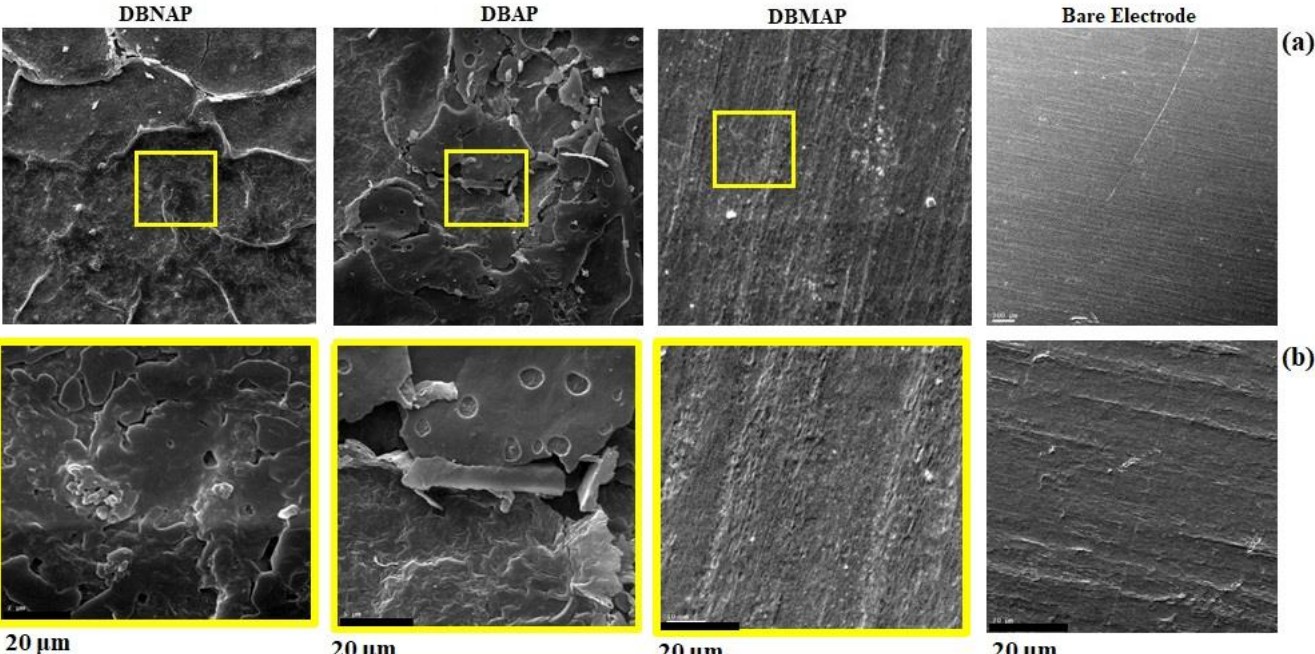

**Figure 11.** Scanning Electron Microscopy images of the surfaces of ion-selective electrodes derived from 4,4-Diformyl(2-aminophenol)dibenzo-18-crown-6 (DBAP), 4,4-Diformyl(2-amino-5-methylphenol)dibenzo-18-crown-6 (DBMAP) and 4,4-Diformyl(2-amino-5-nitrophenol)dibenzo-18-crown-6 (DBNAP) at (**a**) ×775 and (**b**) ×6200 magnification.

However, to assess the usefulness of these electrodes in the detection and quantification of aqueous lead ions, differential pulse anodic stripping voltammetry (DPASV) measurements, a well-tested and highly sensitive method, was employed. Interestingly, whereas the bare electrode reveal two broad bands at −0.8 and −0.4 V, both of which showed no clear trend with lead ion concentration (See Figure S4), for these modified electrodes, a sharp lead peak between −0.60 and −0.50 V which respond linearly to lead ion concentration was observed (Figure 12). Hence, regression analysis was applied to probe the analytical parameters of these electrodes (Table 3). Based on these results, the overall analytical superiority of the DBNAP electrode is indicated by its very wide linear range compared to the other electrodes. This suggests that the DBNAP electrode offers a larger number of interaction sites for $Pb^{2+}$, hence, more analyte is required for the saturation of this electrode. This might be due to the multi-layer structure of the film which allows permeation of the $Pb^{2+}$ where it is able to interact with additional binding sites, especially since the pores and cracks on the film surface (Figure 11) offer such opportunities. Additionally, the presence of high electron density in the $NO_2$ groups presents additional opportunities for binding.

Hence, this electrode offers a wider linear range than that offered by flame atomic absorption spectroscopy (FAAS) which is a very popular and well-regarded method (Table 3). Interestingly, as indicated by the limit of detection (LOD) values, derived from

linear regression [36], all three electrodes are nearly identical where sensitivity is concerned, an indication that the binding sites are probably identical for all electrodes. However, though these values are higher than those for FAAS, the difference in the order of magnitude is not excessive: ∼×10. The same is also true for the limit of quantification (LOQ) values. The such difference are most likely related to the energetics of the analyte-electrode interactions; that is, the stability of the complex formed between $Pb^{2+}$ and the interactions sites, encourage the formation of such interactions. Indeed, this feature can be controlled by functionalisation; that is, altering the binding sites to include binding functions that form stronger/more favourable interactions with $Pb^{2+}$ could improve these LOD and LOQ values. This hypothesis is supported by the fact that polyphenylenediamine and other polymer-derived ISEs, offering nitrogen-containing binding sites have been reported with linear ranges (LRs) spanning the micro and even nano-molar ranges [37–39]. Additionally, an adjustment in the structure of the film on the electrode surface by altering the deposition parameters could also yield changes in the electrode behaviour. Nonetheless, though not as sensitive as FAAS, these electrodes could still find application in the quantification of lead ions. However, if they are to find "real-life" applications, their performance in the presence of other metal ions is critical since environmental and even biological samples are often contaminated with various other metal ions.

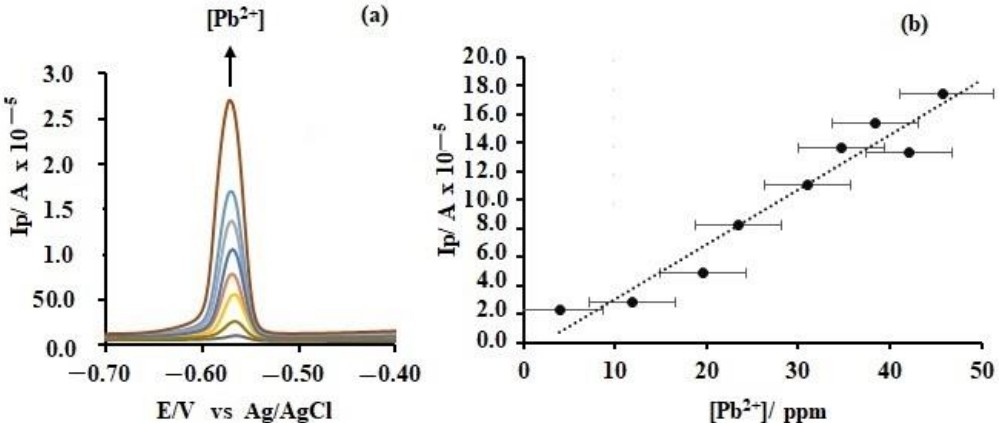

**Figure 12.** (**a**) Typical differential pulse anodic stripping voltammetry (DPASV) curves at different lead ion concentrations, exemplified by 4,4-Diformyl(2-amino-5-methylphenol)dibenzo-18-crown-6 (DBMAP) derived ion-selective electrode, and (**b**) corresponding calibration curve ($R^2$ = 0.95), revealing the linearity of the response of these ISEs to lead.

**Table 3.** Key analytical parameters exhibited by 4,4-Diformyl(2-aminophenol)dibenzo-18-crown-6 (DBAP), 4,4-Diformyl(2-amino-5-methylphenol)dibenzo-18-crown-6 (DBMAP) and 4,4-Diformyl(2-amino-5-nitrophenol)dibenzo-18-crown-6 (DBNAP) derived ion-selective electrodes in comparison to those presented by flame atomic absorption spectroscopy.

| Electrode | LOD [ppm] | LOQ [ppm] | LR [ppm] |
|---|---|---|---|
| DBAP | 7.64 | 23.14 | 15.75–53.03 |
| DBMAP | 9.38 | 28.43 | 3.98–45.80 |
| DBNAP | 8.38 | 24.41 | 3.98–60.15 |
| FAAS | 0.83 | 2.52 | 0.00–20.00 |

Hence, the effects of different metal ions (interferent) on the DPASV signal (current) were investigated when the analyte-interferent concentration ratio was 1:1. Indeed, these are extreme circumstances since environmental concentrations of such interferent species would be much lower under the majority of circumstances. However, assessment under such extreme conditions creates a more accurate picture of the merits and demerits of these ISEs. For DBAP, though changes in the signal are obvious, the order of magnitude of the current remains unchanged in all cases, except for $Al^{3+}$ (Figure 13). This is probably due to

$Al^{3+}$ ions out-competing the softer $Pb^{2+}$ ions for the hard binding sites offered by this film. Similarly, only $Hg^{2+}$ ions lead to fairly significant changes in the signal for the DBNAP electrode. This correlates with the diffused nature/softness of the electron density at the binding sites for this electrode, as indicated by the ESPM, a feature which might become even more pronounced in the polymeric form. Of course, based on this reasoning, it is no surprise that though there are obvious changes in the signal, no specific metal ion leads to a significant signal change for DBMAP; that is, none of the interferent ions result in a change in the order of magnitude of the signal.

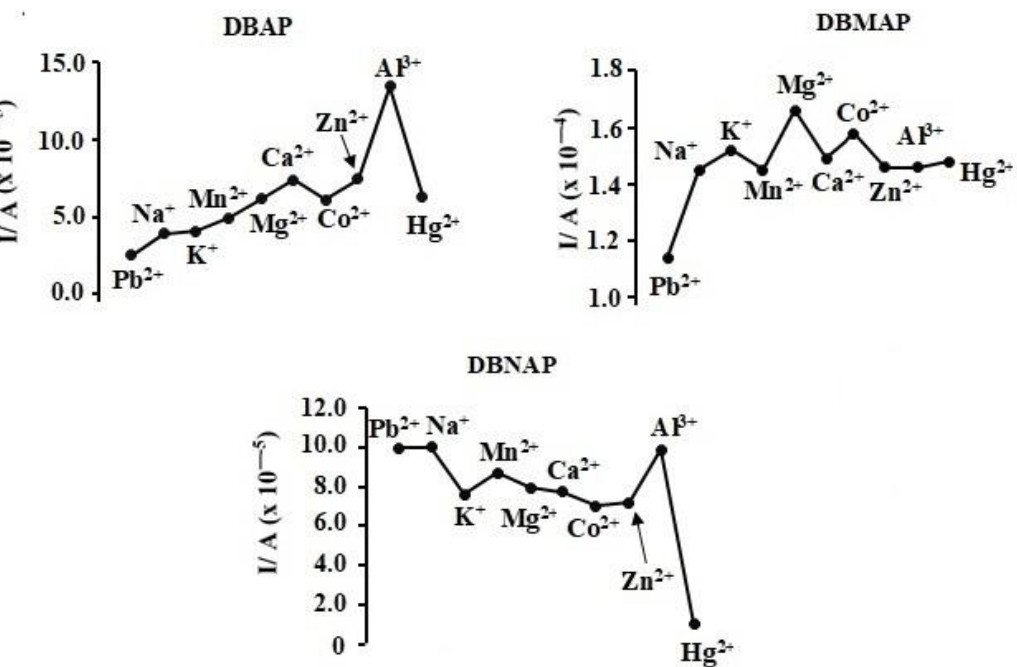

**Figure 13.** Effect of excess (1:1 ratio, $Pb^{2+}$: interferent, $[Pb^{2+}]$ = 100 μM) interfering ions of the differential pulse anodic stripping voltammetry (DPASV) signal generated by 4,4-Diformyl(2-aminophenol)dibenzo-18-crown-6 (DBAP), 4,4-Diformyl(2-amino-5-methylphenol)dibenzo-18-crown-6 (DBMAP) and 4,4-Diformyl(2-amino-5-nitrophenol)dibenzo-18-crown-6 (DBNAP) derived ion-selective electrodes, revealing significant effects of $Al^{3+}$ and $Hg^{2+}$ on their response to lead.

Overall, these results suggest that the electrodes can perform fairly well in the presence of competing metal ions, however, care must be taken with the concentration of interferent metal ions such as $Hg^{2+}$ and $Al^{3+}$; that is, where these metal ions are of similar concentration to lead, a mercury or aluminium error is will occur. Of course, this does not eliminate the application potential of these electrodes since even the highly popular and well-regarded glass membrane electrode suffers from $Na^+$ error. Fortunately, the monomeric units used to create these electrodes can be easily modified, hence, the opportunity for improvement of these ISEs is real.

### 3. Materials and Methods

*3.1. Synthesis*

3.1.1. 4,4-Diformyldibenzo-18-Crown-6

A mixture of dibenzo-18-crown-6 (10 mmol), trifluoroacetic acid (7 mL) and hexamethylenetetramine (10 mmol) was stirred at 90 °C for 12 h, after which the reaction mixture was combined with cold water (~0 °C) with vigorous stirring until an orange-brown precipitate was formed. The precipitated solid was then filtered and washed thrice with water (3 × 10 mL) prior to being recrystallised from ethanol and dried at the pump. A yield of 83% was obtained for this product which melted in the narrow temperature range of 229–232 °C.

### 3.1.2. 4,4-Diformyl(2-Aminophenol)Dibenzo-18-Crown-6 (DBAP)

A mixture of 4,4-diformyldibenzo-18-crown-6 (1 mmol), 2-aminophenol (2 mmol), concentrated hydrochloric acid (1 drop) and dry ethanol (25 mL) was refluxed for 15 h, after which the reaction mixture was cooled to room temperature before being stored on ice until a precipitate was observed. The precipitated solid was filtered whilst being washed with cold ethanol (3 × 5 mL) prior to its recrystallisation from ethanol, after which it was dried at the pump. This product, obtained at a yield of 59.86%, melted in the narrow temperature range of 168–171 °C. The success of this synthesis was confirmed by [1]H, [13]C NMR measurements (see Figure S5a–c) and CH-analysis (see Table S3).

### 3.1.3. 4,4-Diformyl(2-Amino-5-Methylphenol)Dibenzo-18-Crown-6 (DBMAP)

A mixture of 4,4-diformyldibenzo-18-crown-6 (1 mmol), 2-amino-5-methylphenol (2 mmol), concentrated hydrochloric acid (1 drop) and dry ethanol (25 mL) was refluxed for 15 h, after which the reaction mixture was cooled to room temperature and then stored on ice until a solid precipitated from the solution. This solid was then collected by vacuum filtration whilst being washed with cold ethanol (3 × 5 mL), following which it was immediately recrystallised from ethanol and dried, revealing a yield of 20.34%. The resulting product melted in the narrow range of 201–203 °C. The success of this synthesis was confirmed by [1]H, [13]C NMR measurements (see Figure S5b–c) and CH-analysis (see Table S3).

### 3.1.4. 4,4-Diformyl(2-Amino-5-Nitrophenol)Dibenzo-18-Crown-6 (DBNAP)

A mixture of 4,4-diformyldibenzo-18-crown-6, (1 mmol), 2-amino-5-nitrophenol (2 mmol), concentrated hydrochloric acid (1 drop) and dry ethanol (25 mL) was refluxed for 24 h, after which the reaction mixture was cooled to room temperature and then stored on ice until a solid precipitated from the solution. This precipitate was then collected by vacuum filtration with washing (3 × 5 mL cold ethanol), prior to its recrystallisation from ethanol, revealing a final yield of 33.34% and a melting point in the narrow range of 164–166 °C. The success of this synthesis was confirmed by [1]H, [13]C NMR measurements (see Figure S5a–c) and CH-analysis (see Table S3).

### 3.2. Computational Methods

All calculations were conducted via a Gaussian 16 software package [40]. Density functional theoretical calculations were effected via the Pople-style [41] Triply split valence basis set with diffused and polarisation functions on both heavy and hydrogen atoms: 6-311++G(d,p), allowing accurate consideration of proton transfer, long-range interactions and electronic excitation whilst preventing/reducing excess energy over-estimation but remaining efficient (cheap). Fermion exchange was accounted for by Becke's [42] three-parameter hybrid functional whereas Lee, Yang and Parr's [43] model was employed for electron correlation with restriction on fermion spin. Both the coulomb attenuated (RCAM-B3LYP)[44] and non-coulomb attenuated forms (RB3LYP) were explored in this work as outlined in the upcoming sections. These methods, benchmarked by Pereira and co-workers [45], have been shown to be highly accurate in molecular geometry and energy barrier predictions; hence, they were applied herein.

### 3.3. Infrared Spectroscopy

Infrared spectra were recorded from grounded samples, deposited on a germanium window, using a Bruker Tensor 37 FT-IR spectrometer, in the range of 4000–600 cm$^{-1}$, at a resolution of $\pm 1$ cm$^{-1}$.

### 3.4. Absorption Spectroscopy

Absorbance measurements were performed on an HP 8453A diode array spectrophotometer. All solutions were allowed to equilibrate for ca. 2 h. following which their absorbances were recorded within the range of 250–500 nm.

### 3.5. Electrochemical Methods

All electrochemical studies were carried out using a CorrTest CS350 potentiostat/Galvanostat workstation, controlled by the CS studio5 Software package. The applied three-electrode set-up was composed of a silver-silver chloride reference whereas platinum was used for the working and counter electrodes. Platinum electrodes were prepared by washing with aqueous acid followed by deionised water prior to being dried under a stream of dry nitrogen. All potentiostatic EIS measurements were conducted at 10 mV rms (AC) with initial and final frequencies of 100,000 and 0.1 Hz respectively. All measurements were carried out under a static nitrogen atmosphere

## 4. Conclusions

Three novel dibenzo-18-crown-6 imines were successfully synthesised and structurally characterised via various spectroscopic methods. All three compounds are highly asymmetric, structurally and electronically. Nonetheless, they spontaneously form strong coordinate interactions with lead ions, leading to obvious changes in their optical features. All compounds are redox active and are able to be electro-polymerised onto a platinum electrode surface, allowing the formation of lead ISEs which are able to detect lead ions at concentrations as low as 10 ppm even in the presence of high concentrations of other cations, under aqueous conditions. However, these ISEs suffer from mercury and aluminium errors when such metal ions are of similar concentration to lead ions. Nonetheless, they offer great potential for development since they exhibit a wider linear range than Flame Atomic Absorption Spectroscopy, a well-regarded traditional method. Overall, these electrodes hold potential for real-life application. Furthermore, given the relative ease of synthesis and structural modification, the designing/tailoring of their lead interaction characteristics is relatively simple.

**Supplementary Materials:** The following supporting information can be downloaded at: https://www.mdpi.com/article/10.3390/inorganics11070275/s1. Figure S1: (a) Labelled optimized structure of 4, 4-Difomyl (2-amino phenol) dibenzo-18-crown-6 (DBAP), (b) Labelled optimized structure of 4,4-Difomyl(2-amino-5-nitrophenol)dibenzo-18-crown-6 (DBNAP), (c) Labelled optimized structure of 4,4-Difomyl(2-amino-5-methylphenol)dibenzo-18-crown-6 (DBMAP), (d) Solid state Infrared spectra of 4,4-Difomyl(2-amino-5-nitrophenol) dibenzo-18-crown-6 (DBNAP) and its lead complex, collected on powdered samples at ambient temperature, (e) Solid state Infrared spectra of 4,4-Difomyl(2-aminophenol) dibenzo-18-crown-6 (DBAP) and its lead complex, collected on powdered samples at ambient temperature, (f) Solid state Infrared spectra of 4,4-Difomyl(2-amino-5-methylphenol) dibenzo-18-crown-6 (DBMAP) and its lead complex, collected on powdered samples at ambient temperature; Table S1: (a) Bond angles and lengths for 4,4-Difomyl (2-aminophenol)dibenzo-18-crown-6 (DBAP) as calculated using DFT/6-311++G(d,p)/RB3LYP, (b) Bond angles and lengths for 4,4-Difomyl(2-aminophenol)dibenzo-18-crown-6 (DBAP) as calculated using DFT/6-311++G(d,p)/RB3LYP, (c) Bond angles and lengths for 4,4-Difomyl(2-amino-5-methylphenol)dibenzo-18-crown-6 (DBMAP) as calculated using DFT/6-311++G(d,p)/RB3LYP; Table S2: Molecular orbital surface for 4,4-Difomyl(2-amino-5-methylphenol) dibenzo-18-crown-6 (DBMAP) and 4,4-Difomyl(2-amino-5-nitrophenol)dibenzo-18-crown-6 (DBNAP), calculated based on DFT/6-311++G(d,p)/RB3LYP; Figure S2: Calculated Electrostatic potential map surfaces (Isoval. 0.0040) for 4,4-Difomyl (2-aminophenol)dibenzo-18-crown-6 (DBAP) and 4,4-Difomyl(2-amino-5-nitrophenol)dibenzo-18-crown-6 (DBNAP) radical cations, calculated based on DFT/6-311++G(d,p)/RB3LYP; Figure S3: (a) Multi-cycle voltammogram for 4,4-Difomyl(2-amino-5-methylphenol)dibenzo-18-crown-6 (DBMAP), collected during polymerization. Positive (ps) and negative (ns) cyclic voltammetric scans and square wave voltammogram for 4,4-Difomyl(2-amino-5-nitrophenol)dibenzo-18-crown-6 (DBNAP), (b) Multi-cycle (30 scans) voltammogram, collected during polymerization. Positive (ps) and negative (ns) cyclic voltammetric scans and square wave voltammogram for 4,4-Difomyl(2-aminophenol)dibenzo-18-crown-6 (DBAP); Figure S4: Differential pulse anodic stripping voltammetry (DPASV) curves using the bare electrode (platinum) at different lead concentrations (60 μL per-addition, [Pb2+] = 100 μM); Figure S5: (a) Proton and Carbon-13 NMR spectra of 4,4-Difomyl(2-aminophenol)dibenzo-18-crown-6 (DBAP), collected in d6-DMSO, (b) Proton and Carbon-

13 NMR spectra of 4,4-Difomyl(2-amino-5-nitrophenol)dibenzo-18-crown-6 (DBNAP), collected in d6-DMSO, (c) Proton and Carbon-13 NMR spectra of 4,4-Difomyl(2-amino-5-methylphenol)dibenzo-18-crown-6 (DBMAP), collected in d6-DMSO; Table S3: Carbon and hydrogen composition analysis for 4,4-Difomyl 2-aminophenol) dibenzo-18-crown-6 (DBAP), 4,4-Difomyl(2-amino-5-methylphenol)dibenzo-18-crown-6 (DBMAP) and 4,4-Difomyl(2-amino-5-nitrophenol)dibenzo-18-crown-6 (DBNAP).

**Author Contributions:** Conceptualization, manuscript preparation, resources, P.N.N.; Investigation, methodology, data curation, D.T.J.; data curatian, K.W., data curatian, R.A.T. All authors have read and agreed to the published version of the manuscript.

**Funding:** This work was funded by the National Commission on Science and Technology, Jamaica.

**Data Availability Statement:** Data is available on request.

**Conflicts of Interest:** We hereby declare that there are no conflicts of interest and that this work was conducted solely for academic purposes.

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
