# Peer review of "Preparation and Properties of Three Plasticiser-Free Novel Di-benzo-18-Crown-6 Aldimine-Derived Lead(II) Ion-Selective Electrodes"

_inorganics, doi:10.3390/inorganics11070275_

Round 1
Reviewer 1 Report
In this work, three novel dibenzo-18-crown-6 imines have been synthesized and characterized, especially for ion selective electrode (ISE) applications towards lead ion. However, the resutls are not well presented making it very hard to follow.
There is a lack of experimental details, especially electrochemistry section. Which electrolyte is used for Fig. 8?
Mechanism for electropolymerization must be explained and illustrated with the polymer structures.
Apply the surface covergage equation, which is for a surface-controled redox process, for Fig. 8c is not proper, as it’s a diffusion controled process.
The experimental details for EIS must be provided. The fitting to obtain Rct is definitely wrong for DBNAP in Fig. 9a.
The performance of bare electrode without polymerization towards lead should also be presented as a control.
Rationalization of different preformance in Table 3 is not well provided.
Author Response
In this work, three novel dibenzo-18-crown-6 imines have been synthesized and characterized, especially for ion selective electrode (ISE) applications towards lead ion. However, the resutls are not well presented making it very hard to follow.
Authors Reply:
We took some time to reflect on this comment and have made some efforts to improve the readability of our manuscript.
There is a lack of experimental details, especially electrochemistry section. Which electrolyte is used for Fig. 8?
Authors Reply:
Additional details, as suggested, have now been included in the text and the caption of figure 8
We have also added an experimental section for electrochemistry.
Mechanism for electropolymerization must be explained and illustrated with the polymer structures.
Authors Reply:
A mechanism and chemical structure has been proposed an incorporated into the manuscript as suggested. This proposal is based on the DFT and CV data.
Apply the surface covergage equation, which is for a surface-controled redox process, for Fig. 8c is not proper, as it’s a diffusion controled process.
Authors Reply:
Indeed, movement of the material to the surface is diffusion controlled, however, the presence of adsorption waves indicate that the monomers tend to stick to the electrode surface before oxidation, hence, the faradic process is preceded by the adsorption waves. This clearly indicate that the reaction occurs on the surface between surface bound molecules. Infact, this was outlined in our article as quotes there from
“Application of the popular Randles-Ševčík model: ip = 2.69 x 105 n3/2AD1/2cv1/2, reveal that the main oxidation wave is diffusion controlled for all compounds. However, adsorption effects result in partial passivation of the electrode surface on each cycle, hence, cleaning of the electrode surface between cycles results in reduced correlation coefficient (R2 = 0.87 to 0.98) for these plots (ip vs v1/2). Such observations confirm that the pre-oxidation bands, observed on the low energy side of the main oxidation wave, are in-fact adsorption waves, as postulated in the foregoing.”
The experimental details for EIS must be provided. The fitting to obtain Rct is definitely wrong for DBNAP in Fig. 9a.
Authors Reply:
We make the corrections to the RcT values and also included an experimental section for the EIS work.
The performance of bare electrode without polymerization towards lead should also be presented as a control.
Authors Reply:
We have now incorporated information on the behaviour of the bare electrode into the text as a comparison the modified electrode. We have also added, a concentration study for the bare electrode to the supporting information (S8) as evidence of its inability to be used for the sensing and quantification of lead.
Indeed, this was also considered in our study. The bare electrode showed a broad peak that did not react linearly to lead concentration. Secondly, this peak did not show up in the rage being discussed.
Rationalization of different preformance in Table 3 is not well provided.
Authors Reply:
The data in Table was explained from a fundamental perspective. For example, for the DBNAP electrode the following explanation was offered for its wide linear range
“The overall analytical superiority of the DBNAP electrode is indicated by its very wide linear range compared to the other electrodes. This suggests that the DBNAP electrode offers a larger number of interaction sites for Pb2+, hence, more analyte is required for saturation of this electrode. This might be due to the multi-layer structure of the film which allows permeation of the Pb2+ where it is able to interact with additional binding site, especially since the pores and cracks on the film surface (Fig. 11) offer such opportunities. Additionally, the presence of high electron density on the NO2 groups present additional opportunities for binding.”
We believe that this reasoning is accurate since the amount of lead that can be detected is affected by the number of interaction sites that can individually interact with lead and in so doing contribute the current (signal). Of-course, once the lead concentration exceeds the number of interaction sites that can operate independently, the calibration line starts to curve.
For the LOD and LOQ values, we have improved the explanation for the observed values
Reviewer 2 Report
(inorganics-2280866)
The current work provides the fabrication and characterization of three novel dibenzo-18-crown-6 aldimines. Structural characterization of the synthesized compounds was performed via various spectroscopic methods (1H-,13H-NMR, FT-IR), the results are supported by Density Functional Theoretical (DFT) modelling. The aim of this study, declared by the Authors, was to assess the application potential of these imines for the development of modified electrodes which can allow relatively cheap and rapid electrochemical sensing and quantification of aqueous lead ions – in the opinion of the reviewer, this goal was not met.
The first part of the manuscript connected with the structural characterization of compounds is well-prepared. The amount of work performed for this part of manuscript preparation deserve to be appreciated. What is more, this part constitutes the majority of the presented manuscript.
The electrochemical part is short, what is more no constructive conclusions were drawn. In Reviewer opinion based on the obtained electrochemical results it is impossible to determine the analytical usefulness of the developed sensors. From analytical point of view the article was very poorly prepared. Thus, an electrochemical part of the manuscript should be substantially rewritten and filled with conclusions. Otherwise manuscript should be rejected, as presented application potential of the synthesized compounds are very poor.
Following remarks should be taken into account:
1. In Introduction section, Authors should explain in detail why did they choose oxo-crown system and why did they believe that such compound may be selective toward lead ions?
2. Whether such compounds have been used before for analytical purposes? – such discussion should be added in the introduction section as well.
3. Methodology section- no information about methodologies of performed experiments. How did the Authors prepare the platinum electrodes ? What was the procedure of electrode modifications? What kind of equipment did they use?
4. For electrochemical measurement the Authors used acetonitrile, but why? It should be explain… What is more choice of supporting electrolyte is one of the main step in voltammetric experiments, thus should not be omitted.
5. In voltammetry peaks are observed not waves.
6. “low current adsorption wave is also obvious on the low energy side of the main oxidation, at ca. 2.29 V” – this statement is unclear, what the Authors mean by adsorption wave??
7. Randles-Sevcik equation may be used for reversible electrochemical processes.
8. Why did the Authors conclude that the process is diffusion controlled? No sufficient explanation is provided.
9. “The overall analytical superiority of the DBNAP electrode is indicated by its very wide linear range compared to the other electrodes” – in Reviewer opinion the obtained results should be compare with literature data not only with tested in the manuscript electrodes.
10. The obtained calibration curve is not acceptable. Could you explain what did you mean by adding error bars like that??? In is non linear etc. Where is R2?
11. The obtained analytical parameters are not acceptable as well. How did you calculate these values? How it is possible that LOQ value is much higher that the lowest concentration in LR??
12. In my opinion selectivity of the sensors is rather poor, all ions affected the response.
To conclude, the electrochemical part of the manuscript should be substantially rewritten and filled with conclusions. In my opinion, extensive work is necessary to make the paper acceptable for Inorganics. I recommend the major revision of the manuscript.
Author Response
The current work provides the fabrication and characterization of three novel dibenzo-18-crown-6 aldimines. Structural characterization of the synthesized compounds was performed via various spectroscopic methods (1H-,13H-NMR, FT-IR), the results are supported by Density Functional Theoretical (DFT) modelling. The aim of this study, declared by the Authors, was to assess the application potential of these imines for the development of modified electrodes which can allow relatively cheap and rapid electrochemical sensing and quantification of aqueous lead ions – in the opinion of the reviewer, this goal was not met.
Authors Reply:
We respect this opinion but it is clearly not based on our manuscript because our manuscript demonstrated the synthesis of some novel dibenzo-18-crown-6 imines; the structures of which were characterized by various spectroscopic methods. We also show that these compounds were redox active and could be polymerized under the surface of a platinum working electrode. Without the polymeric film, the bare metal (platinum) gives broad peaks for lead which do not reveal a correlating with lead concentration; that is, the bare electrode cannot be used for the quantification of lead. In fact, we have now added DPASV curves (see S9) for the bare electrode in the presence of various lead concentrations, to prove this point. We have also adjusted the text to include this point. However, with the polymeric films of these compounds, a linear relationship between Pb2+ concentration and current is obvious from Fig. 12. Surely, this is clear proof that these compounds can be used to prepare modified electrodes which responds to Pb2+ in predictable manner. Additionally, we established their analytical boundaries by determining their linear ranges, LOD and LOQ values as is typically done in the literature of modified electrodes which spans thousands of publications. In our study, we also made sure to investigate the effects of other ions on the signal of these novel ISEs. The evidence that we were able to (1) prepare the compounds, (2) prepare the ISEs and (3) to use the prepared ISEs for the detection and quantification of lead is provided in our manuscript.
The first part of the manuscript connected with the structural characterization of compounds is well-prepared. The amount of work performed for this part of manuscript preparation deserve to be appreciated. What is more, this part constitutes the majority of the presented manuscript.
The electrochemical part is short, what is more no constructive conclusions were drawn. In Reviewer opinion based on the obtained electrochemical results it is impossible to determine the analytical usefulness of the developed sensors. From analytical point of view the article was very poorly prepared. Thus, an electrochemical part of the manuscript should be substantially rewritten and filled with conclusions. Otherwise manuscript should be rejected, as presented application potential of the synthesized compounds are very poor.
Authors Reply:
We are happy to hear that the reviewer appreciated the early sections of our manuscript. However, for the electrochemistry section of our manuscript, maybe we were not as clear as the reviewer would have liked. Indeed, we have made some adjustment to make our points a bit more vivid. We have also added more information such as the pH at which the analysis was, response times for these ISEs and also a proposed mechanism for the electrochemical transformation at the working electrode. Nonetheless, we find it challenging that the reviewer concluded that this section was “poorly” prepared and does not allow determination of the analytical usefulness of the presented ISE when in fact, the analytical boundaries/ parameters such as linear range, LOD and LOQ values are presented. Such information, by itself (without words), give a significant description of the application potential of any analytical method. Furthermore, we have presented and described data on the behaviour of these ISEs in the present of other metal ions when attempting to detect Pb2+. Hence, we have made conclusions on their short comings in comparisons to FAAS, as is typically done in the literature.
Following remarks should be taken into account:
- In Introduction section, Authors should explain in detail why did they choose oxo-crown system and why did they believe that such compound may be selective toward lead ions?
Authors Reply:
This question was answered in the introduction where we said
“However, oxo-crown systems are of particular interest since they are known to be highly stable and are generally inert. In particular, benzo-crowns offer the possibility being easily tailored along with the ability to interact with metal ions through size binding and other effects [15]”
However, we agree that we could have been more explicit in stating our view on oxo-crowns and also why we think that they could be applied in the titled way. Hence, we have modified the introduction to be more explicit/ direct where our choice of oxo-crowns are concerned.
- Whether such compounds have been used before for analytical purposes? – such discussion should be added in the introduction section as well.
Authors Reply:
This modification was made
- Methodology section- no information about methodologies of performed experiments. How did the Authors prepare the platinum electrodes ? What was the procedure of electrode modifications? What kind of equipment did they use?
Authors Reply:
This modification was made
The procedure for electrode modification was outlined in the manuscript.
“Hence, repetitive voltammetric cycling within the aforementioned scan window was carried out for all compounds, with the aim of effecting electropolymerization onto the WE; as is typically done [36]. The resulting voltammograms, reveal marked changes in the main oxidation wave on each cycle; that is, a clear decrease in current on each cycle, concomitant with noticeable changes in peak potentials, is obvious (Fig. 8 (c), see S6). Additionally, the emergence of new bands indicates that the observed electrode surface passivation is most likely caused by the formation of a polymeric film of the solution phase DB-derivatives, thereon.”
- For electrochemical measurement the Authors used acetonitrile, but why? It should be explain… What is more choice of supporting electrolyte is one of the main step in voltammetric experiments, thus should not be omitted.
Authors Reply:
The supporting has now been named in the manuscript. Additionally, an explanation for the use of acetonitrile has also been added to the manuscript.
- In voltammetry peaks are observed not waves.
Authors Reply:
These modifications were made in the manuscript
- “low current adsorption wave is also obvious on the low energy side of the main oxidation, at ca. 2.29 V” – this statement is unclear, what the Authors mean by adsorption wave??
Authors Reply:
In our article we mentioned that these peaks originated from adsorption processes. Adsorption peaks are observed when the analyte sticks to the electrode surface before an electron transfer. This process does not follow Faraday’s law, hence, they are referred to as non-faradic processes. These peaks are often observed in the literature. For example,
10.1021/acssuschemeng.2c04584
10.3389/fchem.2020.00163
10.1007/s12678-020-00627-6
Since, asked these questions, we decided to insert an explanation into the manuscript to aid reader just in case they might have the same questions.
- Randles-Sevcik equation may be used for reversible electrochemical processes.
Authors Reply:
This not true!
There is a version of the Randles-Sevcik model that applies to quazi-reversible and irreversible systems as presented by Bard, Sevcik and others in the following references:
Randles, J.E.B. A cathode ray polarograph. Part II.—The current-voltage curves. Trans. Faraday Soc. 1948, 44,
327–338.
Ševˇcík, A. Oscillographic polarography with periodical triangular voltage. Collect. Czech. Chem. Commun.
1948, 13, 349–377.
Bard, A.; Faulkner, L. Electrochemical Methods: Fundamentals and Applications; John Wiley & Sons, Inc.:
New York, NY, USA, 2001
Brownson, D.A.C.; Banks, C.E. The Handbook of Graphene Electrochemistry; Springer: London, UK, 2014
For greater clarity, we have modified this part of our manuscript (the equation presented) to be identical to that presented by the aforementioned experts.
- Why did the Authors conclude that the process is diffusion controlled? No sufficient explanation is provided.
Authors Reply:
We have made some modifications to simplify this explanation of how this conclusion was drawn.
However, it must be noted that the explanation for this conclusion was presented in technical terms in relation to the equation as follows:
Application of the popular Randles-Ševčík model:
ip = 2.69 x 105 n3/2AD1/2cv1/2 (Equation has been modified in the new version of the manuscript), reveal that the main oxidation wave is diffusion controlled for all compounds. However, adsorption effects result in partial passivation of the electrode surface on each cycle, hence, cleaning of the electrode surface between cycles results in reduced correlation coefficient (R2 = 0.87 to 0.98) for these plots (ip vs v1/2).
Application of this model is routine in electrochemistry and so is normally explained technical or none at all. This is like calculating moles of a species; it is assumed that the reader have a grasp of such basics. This is a manuscript for scientist in this area a not for elementary school children!
- “The overall analytical superiority of the DBNAP electrode is indicated by its very wide linear range compared to the other electrodes” – in Reviewer opinion the obtained results should be compare with literature data not only with tested in the manuscript electrodes.
Authors Reply:
This adjustment has been made
- The obtained calibration curve is not acceptable. Could you explain what did you mean by adding error bars like that??? In is non linear etc. Where is R2?
The error bars result from errors in the lead solutions that were prepared in order to generate the calibration curve, hence, these error bars are only horizontal. All data of this type will errors. I our case the average error was determined to be ca. ±5 % which is quite good. The use of error bars is most common and useful for such data. Nonetheless, since the reviewer is asking such questions, the R2 value has been incorporated into the figure caption. It I clear that the overall plot shows a direct relationship between peak current (Signal) and lead concentration within a 5 % error margin. It makes greater sense to consider an error margin than to be talking about R2.
- The obtained analytical parameters are not acceptable as well. How did you calculate these values? How it is possible that LOQ value is much higher that the lowest concentration in LR??
Authors Reply:
For these electrodes, the linear regression method was used to determine LOD and LOQ as reported in:
10.4103/2229-5186.79345
and elsewhere. This method is quite standard for linear calibration curves. The way in which the LR was determined is not connected to the methods used to determine LOD and LOQ. The LR as determined here are within ±5% error margin; a treatment used for all three electrodes. This type of treatment is typical in ISE literature.
Furthermore, for ISE, the ideas of LOD, LOQ and LR are not that straight forward. For example, there is a special definition for LOD as posed by the IUPAC then there is another definition which takes into account the performance of the ISE. More information on this is presented in Dillingham’s work (See reference below) and also presented by Radu (see link below).
We have included these perspectives into our manuscript just in case readers might wonder about the same things as the reviewer.
- Dillingham, T. Radu, D. Diamond, A. Radu and C. McGraw, Electroanalysis, 2012, 24, 316324
https://raduresearchgroup.wordpress.com/about/projects/bayesian-calibration-of-ion-selective-electrodes/
- In my opinion selectivity of the sensors is rather poor, all ions affected the response.
To conclude, the electrochemical part of the manuscript should be substantially rewritten and filled with conclusions. In my opinion, extensive work is necessary to make the paper acceptable for Inorganics. I recommend the major revision of the manuscript.
Authors Reply:
We definitely disagree with your conclusion that the selectivity of the sensor is poor because this conclusion was made without considering the fact that the sensors were tested under extreme conditions (1:1 analyte: interferent concentration) instead of taking a under-handed approach, as is common in the literature, and testing the sensors at low interferent concentrations. The fact is, our results are honest. Furthermore, it is not uncommon for an ISE to react to high concetrations of a particular ion. Take for example a pH (Glass membrane ISE); an excess of various ions will cause signal fluctuations!
Indeed, we have made some significant adjustments to the manuscript.
Reviewer 3 Report
This work is quite interesting and important in the field of ion sensing, where they can fabricate the plastisizer-free ISM. The experimental results are also quite comprehensive. A few comments and suggestions can be found below,
- How is the potential of these ionophore in the real application? what is the practical concentration of the interference ions (e.g. mercury (ii), Aluminium (iii). Did you test the interference ions in the proper concentration as the real samples?
- The introduction part lack of description about the current progress and literature regarding the crown ether for lead ion detection. This part seems necessary to be discussed in introduction about the previous work on ionophore for lead ion detection
-The description and detail information of physicochemical and electrochemical characterizations should be added in the methodology part.
-In the caption of each figure, it would be much better to also put the information regarding what kind of measurements or technique were used to acquire the data shown in the respective figure.
- Please improve the quality and clarity of the figure 2, including the position for the title of Y axis should close to the value
- Please modify the figures with proper sub figure division
- For the SEM image, the authors should include the image of bare electrode surface as the control surface.
Author Response
This work is quite interesting and important in the field of ion sensing, where they can fabricate the plastisizer-free ISM. The experimental results are also quite comprehensive. A few comments and suggestions can be found below,
Authors Reply:
We are very happy to know that the reviewer is appreciative of our hard work. We are very grateful for this comment.
- How is the potential of these ionophore in the real application? what is the practical concentration of the interference ions (e.g. mercury (ii), Aluminium (iii). Did you test the interference ions in the proper concentration as the real samples?
Authors Reply:
Under “real world” Al3+ and Hg2+ concetrations will be in the micro- and even the nano-molar range, however, we wanted to test our electrodes under extreme circumstance so the interferent conctrations were set to be 1:1 with the analyte (Pb2+) whose concentration was 100 μM. We anticipate that at lower concentrations, the interferent ions will have a much reduced effects.
Given the importance of these questions posed by the reviewer, the answers, as given here, have also been incorporated into our manuscript.
- The introduction part lack of description about the current progress and literature regarding the crown ether for lead ion detection. This part seems necessary to be discussed in introduction about the previous work on ionophore for lead ion detection
Authors Reply:
This discussion has now been incorporated into the introduction.
-The description and detail information of physicochemical and electrochemical characterizations should be added in the methodology part.
Authors Reply:
This adjustment has been made; that is, a new section has been added to the methodology that gives all the electrochemical parameters and methodologies.
-In the caption of each figure, it would be much better to also put the information regarding what kind of measurements or technique were used to acquire the data shown in the respective figure.
Authors Reply:
All the figure captions have been revised as suggested.
- Please improve the quality and clarity of the figure 2, including the position for the title of Y axis should close to the value
Authors Reply:
These suggested modifications have been made.
- Please modify the figures with proper sub figure division
Authors Reply:
These suggested modifications have been made.
- For the SEM image, the authors should include the image of bare electrode surface as the control surface.
Authors Reply:
These suggested modifications have been made.
Round 2
Reviewer 1 Report
The manuscript has been largely improved.
Author Response
Dear reviewer,
Thank you for your most constructive comments and guidance. We have spent some time revising and rechecking every line of the manuscript and have made some adjustments to the language and flow therein.
Regards,
Peter Nelson, on behave of the other authors.

Reviewer 2 Report
From analytical point of view the presented article is still very poor. In Reviewer opinion the Authors should resign from this analytical part, or rewrite it. In the current state it is not accebtable.
Unfortunately, the authors' answers are not accurate and sufficient.
Author Response
Dear reviewer
We have made significant efforts to improve our article in accordance with your suggestions and also with the other reviewers. So far, the other reviewers have made recognized our efforts. However, you have been insistent on our article been written like an analytical chemistry article when it is obvious that our article is not an analytical chemistry article. Therein, we have presented the development of some ISEs from a fundamental perspective, in keeping with the type of data typically used to make the main deductions on ISEs. The structure, flow, quantity and quality of the data presented therein, is typical for such work. We have painted a clear and honest picture of the merits and demerits of the titled ISEs. In our article, we proposed that these systems could find application as ISEs for the detection and quantification of lead, and we have demonstrated to what extend they would be useful in doing such. Indeed, they are not perfect and we made sure to allow such imperfections to be visible. However, this article represent a foundation for other authors to think and to create better systems; that is, we have done the pure chemistry ground work which can allow others to build-out the applied chemistry aspects in a more holistic manner. Indeed, we will not waste time trying to change your view on our work because it is clear that you are fixated on our article being written like an analytical chemistry article which is not what we are about. The analytical section is just a brief exploration of their possible usefulness in that regard.
On the subject of wrong and right, the reviewer should also acknowledge that he/she made some erroneous comments. For example, when the reviewer commented that the Randles-Sevcik model can not be used to deal with irreversible redox systems, this was straight up wrong and we provided evidence in the form of publications by Bard, Randles and Sevcik themselves. The reviewer also tried to treat ISEs like regular analytical tools without consider for their operating mechanisms.
Cheers,
Peter Nelson on behave of all authors
Round 3
Reviewer 2 Report
Dear Authors,
I assure you that I am not fixated on analytical approach, however if you try to publish some analytical data it should be done with a respect to generally accepted rules for presenting such data.
I appreciate the work that you have done, however I cannot agree with you, and still I stand by my opinion, article in current state is not suitable for journal.
Author Response
As a reviewer, we respect your professional opinion and also thank you for your service. Additionally, we would like to apologize to you if our last set of comments sounded rude because that was honestly not our intent.
We have done some work on the English and typos in the manuscript.
It is also our plan to take onboard your comments on the analytical section as we go forward in our journey to continue working on such systems (ISEs).